# Cortical evoked activity is modulated by the sleep state in a ferret model of tinnitus. A cross-case study

**Linus Milinski\*, Fernando R. Nodal, Matthew K. J. Emmerson, Andrew J. King**[ORCID]**,
Vladyslav V. Vyazovskiy, Victoria M. Bajo**[ORCID]**\***

Department of Physiology, Anatomy and Genetics, University of Oxford, Oxford, United Kingdom

\* linus.milinski@dpag.ox.ac.uk (LM); victoria.bajo@dpag.ox.ac.uk (VMB)

## Abstract

Subjective tinnitus is a phantom auditory perception in the absence of an actual acoustic stimulus that affects 15% of the global population. In humans, tinnitus is often associated with disturbed sleep and, interestingly, there is an overlap between the brain areas involved in tinnitus and regulation of NREM sleep. We used eight adult ferrets exposed to mild noise trauma as an animal model of tinnitus. We assessed the phantom percept using two operant paradigms sensitive to tinnitus, silent gap detection and silence detection, before and, in a subset of animals, up to six months after the mild acoustic trauma. The integrity of the auditory brainstem was assessed over the same period using auditory brainstem response recordings. Following noise overexposure, ferrets developed lasting, frequency–specific impairments in operant behaviour and evoked brainstem activity. To explore the interaction between sleep and tinnitus, in addition to tracking the behavioural markers of noise–induced tinnitus and hearing impairment after noise overexposure, we evaluated sleep–wake architecture and spontaneous and auditory–evoked EEG activity across vigilance states. Behavioural performance and auditory–evoked activity measurements after noise overexposure suggested distinct degrees of tinnitus and hearing impairment between individuals. Animals that developed signs of tinnitus consistently developed sleep impairments, suggesting a link between the emergence of noise–induced hearing loss and/or tinnitus and sleep disruption. However, neural markers of tinnitus were reduced during sleep, suggesting that sleep may transiently mitigate tinnitus. These results reveal the importance of sleep–wake states in tinnitus and suggest that understanding the neurophysiological link between sleep and tinnitus may provide a new angle for research into the causes of phantom percepts and inform future treatments.

## Introduction

Subjective tinnitus, or tinnitus for short, is the most prevalent phantom sensation [1,2]. It is commonly reported as the constant perception of a hissing, ringing or buzzing sound without any identifiable source [1], and is often associated with disturbed sleep [3–11]. During sleep,

**Data Availability Statement:** The data and code supporting the findings of this study are available in the public repository at https://github.com/l-milinski/Code_for_Milinski_et_al_2023.

**Funding:** This work was supported by the Royal National Institute for Deaf People (RNID, Grant S52_Bajo), awarded to VMB, and the Wellcome Trust [WT108369/Z/2015/Z], awarded to AJK. The funders had no role in study design, data collection and analysis, decision to publish, or preparation of the manuscript.

**Competing interests:** The authors have declared that no competing interests exist

the brain is relatively disconnected from the external environment [12–14], neural activity is mainly generated endogenously, and phantom percepts during dreams are considered normal. In contrast, phantom percepts during wakefulness can be a reflection of neuronal malfunction ([15]; work of Ambroise Paré reviewed in [16–18]). Despite the association between sleep and tinnitus, very little is known about the relationship between the neuronal changes that take place during sleep and those that give rise to persistent phantom percepts.

Three main models have been implicated in the development of tinnitus, which attempt to explain the changes that occur in the brain following a cochlear insult: altered lateral inhibition, homeostatic plasticity, and stochastic resonance (reviewed in [19]). In the lateral inhibition model, tinnitus is caused by disinhibition of frequency channels adjacent to deafferented channels (edge effect; [20]); in the homeostatic plasticity model by an increase in spontaneous activity due to an increase in central neuronal gain [21], and, in the stochastic resonance model, by increased noise in a recurrent neural network, leading to subthreshold auditory signals becoming detectable [19].

Tinnitus is associated with functional changes in widely distributed brain regions, including both auditory and non–auditory areas [22–28], many of which exhibit a dramatic modulation in their spatiotemporal activity across vigilance states: for example, wakefulness is dominated by high-frequency, low-amplitude oscillatory brain activity, whereas sleep is dominated by slower, high-amplitude oscillations [29–31]. We recently proposed that the spatial overlap between brain areas affected by tinnitus and those showing sleep–state dependent neural activity may lead to competition between pathological and physiological drives determining cortical network activity [32]. If tinnitus–related activity persists across vigilance states, it may result in a state of partial arousal during sleep similar to that observed in some forms of insomnia and parasomnias [33–35], where the emergence of global and local activation interferes with natural sleep–wake dynamics, potentially causing sleep impairments. Yet, the possibility remains that local and global changes in brain activity across vigilance states [30,36] may, in turn, interfere with tinnitus–related activity. In particular, high–intensity sleep with high levels of cortical slow wave activity, prompted by cellular and network–level drives for recovery sleep [37], such as after a period of extended wakefulness [30,38,39], could potentially mitigate tinnitus temporarily. This leads to the intriguing hypothesis that a dynamic modulation of the phantom sound sensation occurs across the sleep–wake cycle, depending on the relative weighting of circadian and homeostatic drives.

The main aim of this study was twofold: first, to introduce a new animal model for assessing the interaction between sleep and chronic tinnitus, and to validate this model by examining sleep alterations following tinnitus induction; and second, to gather preliminary data to empirically test for a bidirectional relationship between tinnitus and sleep. The ferret is an attractive animal model that, due to its long lifespan, enables longitudinal studies to be carried out. Ferrets further allow for detailed assessment of operant behaviour, and are a valuable model for hearing research in particular, with a hearing frequency range more similar to the human range than in case of rodents. We tracked behavioural markers of noise–induced tinnitus and hearing impairment in a ferret model at one week and six months after noise overexposure and, in parallel, assessed the sleep–wake pattern, as well as spontaneous and auditory–evoked EEG activity across vigilance states. In a similar manner to polysomnography, which is based on electromyogram (EMG) recordings and EEG recordings of global brain activity to assess human sleep and sleep disorders [4,29,40], we used EEG and EMG recordings for vigilance state scoring as previously established in rodents (e.g. [14,41]) and ferrets [42].

From the initial group of eight ferrets exposed to noise and tested in behavioural paradigms, three ferrets were implanted with recording electrodes and tested again six months after noise overexposure. Behavioural performance and auditory–evoked activity suggested distinct

differences in the degree of tinnitus and hearing impairment between individuals. While all implanted animals showed signs of hearing loss, those developing the highest levels of tinnitus severity also exhibited sleep impairments, suggesting that the emergence of noise–induced hearing loss and tinnitus is associated with sleep disruption. Finally, neural markers of tinnitus were reduced during sleep, suggesting that sleep may transiently mitigate tinnitus. Overall, these results highlight the potential of measuring natural brain state dynamics to investigate tinnitus and uncover new avenues for future treatments.

## Material and methods

### Animals

All procedures were carried out in accordance with the Animal (Scientific Procedures) Act 1986 (amended in 2012) and authorised by a UK Home Office Project Licence following approval by the Committee on Animal Care and Ethical Review of the University of Oxford. Adult female pigmented ferrets (*Mustela putorius furo*) were used in this study. Female ferrets are considered more suitable for neuroscientific studies than males because the brain size is similar across adult individuals and the thinner skull and temporalis muscles allow cranial implants to be attached more easily [43–45].

A total of nine ferrets were used in this study. Eight animals were used for tinnitus assessment in the baseline condition and then exposed to noise. One animal was subsequently excluded from the experiment due to health issues unrelated to the procedure and no behavioural data obtained in this animal after NOE were included in the analysis. A subset of four animals were implanted for chronic recordings but only three were used for long-term assessment. Two animals were assessed in all paradigms, whereas one was only used for ABR and operant gap detection measurements (since the experiment had to be terminated prematurely due to health reasons unrelated to the experimental protocol). The animals were chronically implanted with cortical EEG and EMG electrodes to assess sleep–wake architecture and spatio-temporal patterns of brain activity before and after noise overexposure. The longitudinal design of this study spanning more than 9 months allowed each animal to act as its own control by comparing the results before and after noise overexposure. We did not use specific controls comprising implanted animals without noise overexposure, since no changes in behaviour or neural activity were found in previous ferret experiments following similar surgical procedures and/or intracranial implantation [46,47]. One additional naïve animal was used as a control to compare physiological correlates of tinnitus with two out of the three ferrets implanted for chronic recordings.

Animals were housed in small groups of at least three ferrets in standard laboratory enclosures or large pens housing up to ten animals. Food and water were available *ad libitum* except during periods of operant behavioural testing, where water access was mainly limited to rewards received during behavioural testing. Periods of water access regulation lasted for a maximum of five consecutive days before at least two days of *ad libitum* access to water. While under water regulation, animals received performance–dependent amounts of water during the behavioural task, topped up in the form of mashed food puree to an amount of 60 ml/kg body weight per day, to ensure that animals maintained a body weight $\geq$ 85% of their free–feeding weight.

Animals were housed under a 15/9h light–dark cycle from mid–March until mid–November (summer light cycle) and otherwise under an 8/16 h light–dark cycle (winter light cycle) to mimic natural seasonal changes in light exposure [48]. To suppress oestrus, ferrets were routinely injected with Delvosteron (MSD Animal health, Proligestone 100mg/ml) one month before the change to the summer light cycle. Behavioural experiments were conducted

predominantly during the light phase, and chronic EEG recordings took place only during the summer light cycle.

During the periods of continuous EEG recordings (each lasting 2–3 days), animals were single housed in a custom–made enclosure (LxWxH 60x60x70cm) within a double–walled sound–attenuated chamber. Animals had *ad libitum* access to water and food. Identical bedding and nesting material to their home cages were provided, and the lighting and temperature conditions were the same as in the home enclosure. Before the first chronic recordings, ferrets were progressively habituated to the recording enclosure and to the tethering used for EEG recording. Recording enclosures were cleaned after each recording period.

Body weight, fur appearance and social interactions were monitored weekly over the course of the experiment to exclude any general change in animal behaviour due to the surgical or noise overexposure procedure or the recording paradigms.

At the end of the study, final recordings in the auditory cortex under general anaesthesia were conducted in two of the implanted ferrets plus one control case. At the end of the recording sessions, animals were euthanized using pentobarbital overdose (Euthatal, 2 ml of 200mg/ml of Pentobarbital sodium, i.p., Merial Animal Health), which was administered while the animals were still under anaesthesia. All other animals were euthanised at the end of the study using pentobarbital overdose after previous sedation with Domitor (0.1 mg/kg body weight intramuscularly of medetomidine hydrochloride, i.m. Pfizer Ltd).

### Electrode implantation

The aseptic surgical procedure largely followed the methodology previously described in [44,46]. Anaesthesia was induced with a single intramuscular injection of medetomidine hydrochloride (0.022 mg/kg BW; Domitor, Orion Pharma) and ketamine hydrochloride (5 mg/kg; Narketan10, Vetoquinol). Anaesthesia was maintained using Isofluorane (1–3%) (Iso-Flo, Abbot Laboratories) with 100% oxygen as a carrier. Atropine sulphate (0.06 mg/kg, s.c.; Atrocare, Animalcare) was administered to minimise pulmonary secretions along with dexamethasone (0.5 mg/kg, s.c.; Dexadreson, Intervet) to prevent cerebral oedema. Doxapram hydrochloride (4 mg/kg, s.c.; Dopram–V Injection, Pfizer) was administered to minimise respiratory depression. Perioperative analgesia was provided with buprenorphine hydrochloride (0.01 mg/kg, s.c.; Vetergesic, Sogeval) and meloxicam (0.2 mg/kg, s.c.; Metacam, Boehringer Ingelheim). Prophylactic antibiotics to prevent infections were administered during the surgery (Augmentin: co–amyxoclav 0.02 mg/kg i.v. every 2 hours; Bechaam) and once daily for five days after surgery (Xynulox: Amoxicillin trihydrate/Co–amixoclav, 0.1 mg/kg i.m.; Zoetis). Depth of anaesthesia, respiratory rate, ECG, and end–tidal $CO_2$ were monitored and maintained throughout the experiment. The animal's temperature was monitored using a rectal probe and maintained at 38°C using a homeothermic electrical blanket (Harvard Apparatus) and a forced–air warming system (Bair Hugger, 3M Health Care).

The ferret was placed in a stereotaxic frame, the eyes were protected with a carbomer liquid eye gel (Viscotears, Alcon Laboratories), and the skull was exposed. Custom–made wired headmounts (Pinnacle Technology Inc. Lawrence) for EEG recordings were attached to bone–anchored stainless steel screws in contact with the *dura mater*. They acted as EEG electrodes, which were positioned unilaterally over the right frontal (AP 4 mm, ML 4 mm) and occipital (AP 7mm, ML 5mm) cortical areas and over the cerebellum reference electrode). Similar configurations using frontal and occipital of EEG electrodes, optimal for recording cortical slow-wave and theta activity, have been used to provide recordings suitable for vigilance state scoring across different mammalian species such as ferrets, rats and mice [36,42,49]. Two tip–blunted stainless–steel wires were placed into the nuchal muscle for electromyography (EMG).

Wires and screws were secured to the skull surface and protected by covering with bone cement (CMW1 Bone Cement, DePuy CMW, Lancashire, UK). EEG head mounts were protected with accessible plastic enclosures secured to the bone cement. The temporal muscle was temporarily detached at the dorsal part to provide access to the skull so that the EEG electrodes could be fixed to it. At the end of the surgery and to restore its function, the muscle was repositioned over the low profile most lateral part of the cranial pedestal using resorbable sutures and covered with the skin that was sutured independently around the most medial externalised part of the cranial implant. To expedite recovery from anaesthesia at the end of the procedure, animals received Antisedan (atipamezole hydrochloride, 0.06mg/kg, s.c., Vetoquinol). A minimum two–week postsurgical recovery period was allowed prior to further procedures.

## Noise overexposure

In line with previous work in which noise exposure was used to induce tinnitus in animal models, noise presented monaurally in order to prevent hearing loss in both ears [50–52]. Noise (one octave narrowband noise centred at 8 kHz, 98 dB SPL at ear level) was presented for 120 minutes unilaterally via an earphone (Sennheiser CX300 II earphone) attached by a silicone tube to the entrance of the right ear canal while the left ear was fitted with an earplug and silicone impression material (Otoform, Dreve Otoplastik) to minimise its sound exposure. Closed–field calibrations of the sound–delivery system were performed using an 1/8th–in condenser microphone (Brüel and Kjær) attached to the silicone tube. The procedure was carried under general anaesthesia (assessed by immobility and absence of the pedal reflex), which was induced through intramuscular injection of medetomidine hydrochloride (0.022 mg/kg; Domitor, Orion Pharma) and ketamine hydrochloride (5 mg/kg; Narketan10, Vetoquinol). Depth of anaesthesia and the respiratory rate were monitored throughout the procedure. Anaesthesia was maintained by injection of half of the initial dose after 60 minutes or when the animal showed signs of arousal. Body temperature was maintained at 38˚C using a homeothermic monitoring system (Harvard apparatus). To reverse the effect of Domitor and expedite recovery from anaesthesia, animals received Antisedan (atipamezole hydrochloride, 0.06mg/kg, s.c.). Noise overexposure (NOE) took place during the light phase, and animals were given at least 48h of rest before any other procedure.

## Auditory brainstem response measurements

Auditory brainstem responses (ABRs) were obtained under anaesthesia (medetomidine/ ketamine as described in 'Noise overexposure', see above) using sterile subcutaneous monopolar needle electrodes (0.35 x 12 mm, MN3512P150, Spes Medica). Body temperature was maintained at 38˚C using a forced–air warming system (Bair Hugger, 3M Health Care). Stimuli were presented monaurally (left and right in subsequent recordings) via earphones (Sennheiser CX300 II) inserted into the ear canal and fixed in place with silicone impression material (Otoform, Dreve). Auditory stimuli were generated using an RP2.1 Enhanced Real–time processor (Tucker Davies Technologies, TDT) with a sampling frequency of 100 kHz connected to a TDT PA5 programmable attenuator. The earphones were calibrated using SigCalRP TDT calibration software to generate compensation filters ensuring stable levels across a frequency range from 250 to 30,000 Hz. Click stimuli (rarefaction click trains, rectangular voltage pulse, 100 μV, low–pass filtered) were presented at a rate of 17/sec for 700 repetitions per level (40, 50, 60, 70, 80, 90 dB SPL). One octave narrow–band noise stimuli (NBN, centred around 1, 4, 8 and 16 kHz) with a 5 ms duration were presented at a rate of 21/sec for 700 repetitions per level–frequency combination. Signals were recorded from two active subcutaneous electrodes, placed close to the left and right auditory *bullae*, respectively, and referenced to an electrode

placed at the vertex of the skull. A ground electrode was placed on the back of the animal. The signal was routed to a low impedance preamplifier (TDT RA16PA) and headstage (TDT RA4LI) and recorded by an RZ2 Bioamp Processor (25 kHz sampling rate) controlled by Bio-SigRP software (TDT).

## ABR signal analysis

ABR thresholds were determined manually by an experienced experimenter through visual assessment of ABR traces. This was conducted under blind conditions (enabled through a ran-domisation process used to access the data) with respect to animal, stimulus and stage of the experimental timeline (baseline, one week after noise overexposure (NOE), and six months after NOE. Thresholds were defined as the lowest stimulus level where an ABR wave was present if corresponding waves were also present at higher sound levels. If no ABR wave was present for any sound level, the threshold was defined to be at 90 dB SPL (the highest sound intensity used).

Data analysis was performed offline based on the average ABR signals (averaged over 700 individual ABR traces) for each stimulus type. As a readout for the magnitude of the entire ABR signal across all waves, the root mean square (RMS) of the signal was calculated by apply-ing the MATLAB function *rms* on the signal in the predefined response window, 1.6 ms to 4 ms, to include only the ABR signal. To account for longer response latencies at low sound lev-els, the response window was shifted by 0.16 ms for each 10 dB decrement. A level–response plot was computed for each animal, assessment and stimulus and the area under the graph was calculated.

## Operant silent gap detection

Ferrets were trained by operant positive reinforcement using water as a reward to carry out a silent gap–detection task in an arena as described in previous work [53,54] (Fig 1A–1D). The setup consisted of a circular arena (radius, 75 cm) housed in a double–walled sound–attenu-ated room. Animals were trained to initiate a trial by licking a spout, which activated infrared sensors on a platform at the centre of the arena. This ensured that the animal was facing the loudspeaker location at 0˚ azimuth at the time of sound delivery. Licking the central spout trig-gered the presentation of one of two types of sound stimuli through a single loudspeaker (Audax TW025MO). The two stimulus types were either a continuous sound or the same sound including four silent gaps. Following stimulus presentation, the animal had to leave the central platform and approach a peripheral response location at 30˚ to the left in 'gap trials' and 30˚ to the right in 'no-gap trials' to obtain a water reward. Both types of stimuli were pseudo randomly balanced to avoid response bias to either location. There was no time limit for the animals to respond. Incorrect responses were not rewarded. After an incorrect response, trial initiation triggered the identical sound stimulus up to two more times (correc-tion trials) before a new stimulus was presented. Correction trials were not included in the data analysis.

Sound stimuli were generated by TDT System III hardware. The paradigm was controlled by a custom MATLAB program that registered the position of the ferret at the arena centre and response locations, presented the stimuli and delivered the rewards accordingly. Sound stimuli were broadband Gaussian noise bursts (BBN, 30 kHz lowpass) and one octave narrow-band noise bursts (NBN) centred at 1, 4, 8, and 16 kHz. In gap trials, four equally spaced, iden-tical silent gaps were introduced in the stimulus. Across trials, the length of these gaps varied from 3, 5, 10, 20, 50, 100 to 270 ms in duration. Stimuli were generated *de novo* for each trial, cosine ramped with a 10 ms rise/fall time and had a total duration of 2080 ms. All stimuli were

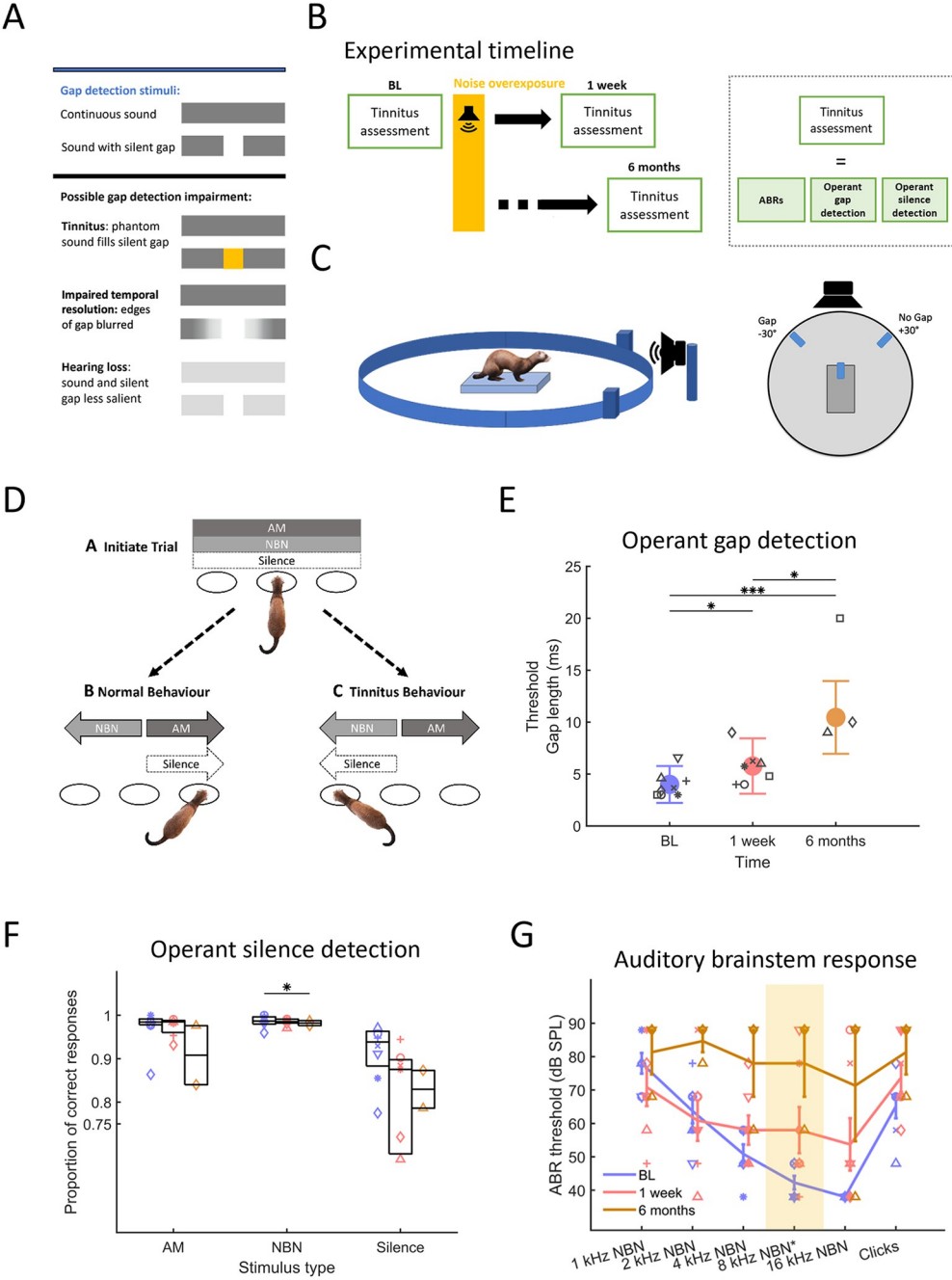

**Fig 1. Behaviour and auditory brainstem responses are impaired after noise overexposure.** (**A**) Stimuli used in gap-in-noise detection paradigms. Different hypotheses on how silence gap detection could be affected by tinnitus filling the gap, degradation of temporal resolution that would blur the edges of the gap and by hearing loss reducing the silent gap saliency. (**B**) Experimental timeline. Animals were assessed in behavioural paradigms (operant silent gap detection, silence detection), and auditory brainstem responses (ABRs) were obtained. Assessments of all these metrics took place once under baseline conditions and on two occasions after noise overexposure (NOE), starting one week after NOE and within two months and six months following NOE. ABRs were always tested on one day in the first week following NOE and again 6 months later, whereas it took two months to complete behavioural testing at each of the three time points. (**C**) Operant gap detection setup. Behavioural arena with a central platform from which the animal initiates a trial, loudspeaker located at 0˚ azimuth relative to the centre platform and water spouts at +30˚ and -30˚. (**D**) Silence detection paradigm. The task took place in the same arena as shown in **C**. Animals were trained to activate the +30˚ sensor for AM and silence trials and the -30˚ sensor for NBN trials. An animal experiencing tinnitus is expected to mistake silence trials for NBN trials and respond accordingly (behavioural task modified from [55]). (**E**) Operant gap

detection. Gap detection thresholds before NOE (BL), 1 week after (1 week) and 6 months after (6 months). Error bars represent standard deviation (SD). Symbols represent individual animals (n = 8 baseline, n = 7 one week, n = 3 six months). Colour coding in each panel and in subsequent figures: Blue for baseline, red for one week, and orange for six months following noise overexposure. (**F**) Operant silence detection. Proportion of correct trials for AM, NBN and silence trials. Error bars represent box plot interquartile ranges across sessions. Asterisk represents statistical significance at p<0.05 between proportion of correct responses at baseline and at six months following noise overexposure. (**G**) Auditory brainstem response (ABR) thresholds, defined as the lowest intensity where a significant response–local peaks of waves 1–4 –could be observed by a trained experimenter under blind conditions. Markers in panels **E**, **F**, **G** represent the mean values from individual animals. Asterisks for panels **E** and **F** represent p values *<0.05, **<0.01, ***<0.001. Experimental timepoints are depicted in blue (BL), red (1 week), orange (6 months).

filtered using the inverse transfer function of the loudspeaker to obtain stable sound intensity levels across the presented frequencies at 76±5 dB SPL.

Animals were tested twice daily in blocks of five consecutive days separated by at least two days of *ad libitum* access to water. Within each session, gap lengths were randomised across gap trials. Stimulus centre frequencies were identical across trials within a given session but varied between sessions to obtain approximately equal numbers of trials for all stimuli (1, 4, 8 and 16 kHz NBN and BBN). Procedural training, not included in the analysis, was provided using only the longest gap length (270 ms) until the animals reached ≥80% correct responses in two consecutive sessions, after which they were tested using all gap lengths.

Animals had to complete 600–1000 trials for each stimulus type. Analysis was based on the average performance for each session. Trials with response times of >5 seconds were excluded from further analysis. Sessions with few trials (more than 5 gap lengths each with <5 presentations) were also excluded from the analysis.

Since a constant phantom sound can fill a silent gap in a presented sound stimulus (S1A Fig), tinnitus may create a bias towards detecting non-gap sounds. Consequently, lower false alarms (FA) and hit rates may occur with tinnitus, effectively compensating for each other when *d'* is calculated. Therefore, we used hit rate or the proportion of correct gap responses rather than calculating *d'* to quantify behavioural performance. Main effects on hit rate were estimated by fitting a GLMM (target distribution: normal, link: identity) on approximately normally-distributed hit rate data (repeated measures: gap length, testing session, stimulus frequency). Note that the number of gap and no-gap trials was equal to prevent animals from developing a bias to one side in the paradigm. Therefore, no-gap and gap trials contributed equally to statistical analysis, and the contribution of each gap trial was equal.

For analysis of silent gap detection thresholds, a sigmoid function (*R P (2022). sigm_fit (mathworks.com/matlabcentral/fileexchange/42641–sigm_fit, MATLAB Central File Exchange)* was fitted on hit rate (between 0 and 1) vs gap length, calculated for each animal, stimulus (1, 4, 8 and 16 kHz NBN and BBN) and condition (baseline, 1 week, and 6 months post NOE). Thresholds were defined as the gap length in closest proximity to a hit rate of 0.5 (chance level) on the fitted function. Fits with slopes at threshold of more than 10 times the median slope across all samples were excluded from the analysis (this applied to 2 out of 78 samples).

## Operant silence detection

Operant silence detection (modified from [55]) took place in the same testing arena as the silent gap detection paradigm, and trials were initiated in the same way. In the silence detection task, however, trial initiation triggered a light emitting diode facing the central platform (signalling the start of the trial) and one of three sound stimulus types: narrowband noise (NBN, one octave bandwidth with centre frequencies at 1, 4, 8, and 16 kHz randomised across trials), a sinusoidally–amplitude modulated BBN (AM stimulus, 100% modulation depth, 5 Hz modulation frequency) or silence (no sound). The proportions of trials in which these stimuli

were presented were 50% for NBN, 30% for AM, and 20% for silence to ensure equal probability of reward in accordance with the following criteria. In AM and silence trials, responses to a sensor located at +30˚ relative to the central platform were rewarded with water, whereas in NBN trials responses to the –30˚ sensor were rewarded. There was no time limit for the animals to respond and a trial was completed whenever the animal responded to either of the two sensors.

During the training phase for this paradigm, reward probability for correct responses was gradually reduced from 1 to 0.7. After animals reached the performance criterion (>80% correct in two consecutive sessions), the paradigm was altered in that silence trials were never rewarded ('testing phase' of the paradigm) to measure the animals' performance without a further training effect. To keep the overall reward probability consistent with the training phase, the reward probability for non–silence trials was 0.9.

The animals completed a total of 1000 trials in this paradigm for each of the testing blocks, baseline and the two assessment blocks at different intervals after noise overexposure (1 week and 6 months). To reacquaint the animals with the paradigm before each testing block, they undertook the training paradigm again until reaching the performance criterion. While this retraining might have allowed animals to adjust to a new 'perceptual baseline' after noise overexposure and therefore mask subtle effects of tinnitus or hearing impairment on performance, it ensured that any variations in performance over time were unlikely to be due to, e.g., the animal forgetting aspects of the paradigm, rather than an effect of NOE. To assess performance, hit rates were compared across stimuli and conditions (Baseline, 1 week and 6 months). Sessions in which an animal completed less than 5 trials for one or more stimulus types were excluded from further analysis. Trials with long response times (>20s) were also excluded.

## Definition of indices for tinnitus and for changes in auditory brainstem responses

As a summary index for behavioural evidence for tinnitus a behavioural tinnitus index (TI, see Eq(1) below) was defined. The TI was calculated as the sum of three metrics obtained as indicators of tinnitus in the two operant tasks, by comparing the values in the baseline condition *(BL)* and Post NOE *(Post)*. Two of the metrics were based on the operant gap detection task (*M(cont)* and *M(thresh)*) and one on the silent detection task (*M*(silence)).

For each metric, M(silence), M(cont), and M(thresh), positive values describe changes in line with tinnitus development (an impairment in silence detection, an increase in continuous sound detection and an increase in gap detection threshold, respectively). The behavioural tinnitus index (TI, see Eq(1)) is the sum of these metrics (Eqs (2–4)), enabling the level of tinnitus and hearing loss experienced by each animal to be parametrised after NOE.

$$TI = M(cont) + M(thresh) + M(silence) \qquad \text{Eq(1)}$$

The metric for operant gap detection performance refers to the change in ability to detect the continuous sound (no gap) across all tested stimulus frequencies (Eq(2)) and to the change in gap–detection threshold across all stimuli (Eq(3)). Eq(2) is based on the assumption that an animal experiencing a continuous phantom sound (tinnitus) should also make fewer errors in detecting continuous sounds due to this constant percept and therefore show improved continuous sound detection ability as compared to before the tinnitus trigger.

$$M(cont) = \frac{Post \text{ \% correct}}{BL \text{ \% correct}} - 1 \qquad \text{Eq(2)}$$

The metric for a change in gap detection threshold (Eq(2)) was based on a normalised

performance value instead of thresholds corresponding to different gap lengths. To establish a normalised threshold, each gap length (3, 5, 10, 20, 50, 100, 270ms) was assigned a performance value based on the assumption that a threshold at 3ms corresponds to 100% performance and the remaining gap lengths correspond to evenly spaced decrements in threshold (85.71%, 71.43%, 57.14%, 42.86%, 14.29%). This approach for defining threshold changes results in a metric reflecting the direction of threshold change and its magnitude on a scale between 0 and 1).

$$M(thresh) = 1 - \frac{Post\ threshold(normlised)}{BL\ threshold(normalised)} \qquad \text{Eq(3)}$$

The metric for operant silence detection Eq(4) refers to the change in silence detection ability (percent correct in silence trials) after NOE.

$$M(silence) = 1 - \frac{Post\ \%\ correct}{BL\ \%\ correct} \qquad \text{Eq(4)}$$

Thus, positive values describe the magnitude of a decrease in silence detection ability (and therefore evidence for tinnitus) and negative values describe the magnitude of an increase in silence detection ability. For example, a value of M (silence) = 0.2 would indicate a decrease in performance by 20% relative to baseline performance. The magnitude is between 0 and 1, the same scale as in the other defined metrics of the tinnitus index (Eqs(2–3)).

**Index for changes in ABRs.** Changes in ABRs were summarised using two metrics: first, changes in ABR thresholds relative to baseline (BL) and second, changes in ABR total magnitude relative to BL.

The metric for thresholds refers to the average change in thresholds across all tested stimuli (in dB), defined as the difference between BL and Post NOE values:

$$M(Thresholds) = (avrg.threshold\ BL - avrg.threshold\ Post) \qquad \text{Eq(5)}$$

The metric for the ABR magnitude is defined as follows: first, the magnitude of the entire ABR was defined as the root mean square of the signal in a predefined response window (1.6-4ms, with a shift of 0.16ms for every decreasing step in sound level) following stimulus presentation. The total magnitude was then calculated for each animal and each stimulus by measuring the area under the level–response graph. The input 'ABRmag' used for the metric below is the average over total ABR magnitudes for all stimuli (1, 2, 4, 8, 16 kHz NBN and BBN) per animal. The metric for each animal is defined as

$$M(ABR\ magnitude) = \left( \frac{Post\ ABRmag}{BL\ ABRmag} - 1 \right) x\ 100 \qquad \text{Eq(6)}$$

Positive values indicate the magnitude of a response increase (in %) whereas negative values indicate a reduced response (evidence for hearing loss).

## EEG signal processing and vigilance state scoring

Data acquisition was performed using a multichannel neurophysiology recording system (TDT). Cortical EEG was recorded from frontal and occipital derivations. EEG/EMG data were filtered between 0.1 and 100 Hz, amplified (PZ2 preamplifier, TDT) and stored on a local computer at a sampling rate of 1017 Hz, and subsequently resampled offline at 256 Hz. Signal conversion was performed using custom–written MATLAB (The MathWorks Inc.) scripts. Signals were then transformed into European Data Format (EDF). Vigilance state scoring was performed manually offline prior to spectral analysis and assessment of sleep architecture. For

data visualisation the EEG was additionally filtered between 0.5 and 30 Hz, and the EMG between 10 and 49 Hz with an additional 50 Hz notch filter to ensure attenuation of potential electrical current noise.

Manual vigilance state scoring was based on visual inspection of consecutive 4-s epochs of filtered EEG and EMG signals (SleepSign, Kissei Comtec Co). Frontal and occipital EEG and neck muscle EMG channels were displayed simultaneously to aid manual scoring and video recordings of the animal were consulted for further validation. Vigilance states were classified as waking (low–voltage desynchronised EEG with high level EMG activity), NREM sleep (presence of EEG slow waves, characterised by high amplitude and low frequency EEG, 0.5–4 Hz), REM sleep (low–voltage, mid–frequency EEG, 4.5–8 Hz, with a low level of EMG activity) or REM2 (low–voltage, high–frequency EEG, 8.5–20 Hz, with a low level of EMG activity. Epochs containing an EEG signal contaminated by artefacts (such as due to gross movements of the animal, eating or drinking) were excluded from subsequent analysis.

For each 24 h recording period, EEG power spectra were computed by a fast Fourier transform (FFT) routine for 4-s epochs (Hanning window), with a 0.25-Hz resolution (SleepSign Kissei Comtec Co). Artefacts in specific frequency bins that had remained unnoticed during manual data scoring were excluded offline. For each 0.25 Hz frequency bin of a given vigilance state during a 24 h recording period, the mean and standard deviation across all epochs was calculated based on a 500-iteration bootstrap. Values outside the mean ± 600 standard deviations across all epochs were excluded from the analysis (to exclude only extreme outliers). Exclusion in this case applied just to the specific frequency bin identified as an outlier. The objective of this was to prevent extreme outliers from distorting the spectral power estimate in particular frequency bins.

## Sound presentation during sleep and wakefulness

Animals were housed individually in a custom–made recording chamber on a 15/9 h light–dark cycle (summer cycle) as described above for the continuous EEG recordings. Frontal and occipital EEG and neck–muscle EMG recordings were obtained over approximately 48–72 hours (2–3 days) per recording session.

In the first 24 h per recording session, the animal was left undisturbed. Over the course of the second 24 h of the session, auditory stimuli were presented via a free–field loudspeaker installed on the ceiling of the double–walled sound–attenuated recording chamber above the custom–made enclosure. Sounds were one octave narrowband stimuli with centre frequencies of 1, 4, 8 and 16 kHz. Stimuli had a duration of 820 ms and included a silent gap of 38 ms in the middle of the stimulus. Stimuli were presented at 40, 50, 60 and 65 dB SPL (as measured at floor level at the centre of the enclosure, where the bedding material in which the ferret sleeps was located). Inter–stimulus intervals had a random duration ranging from 10 to 42 s and each stimulus–level combination was presented 200 times. Stimuli were generated via MATLAB and produced using an RP2.1 Enhanced Real–Time Processor (TDT) and an Alesis RA150 Amplifier. Stimulus presentation was controlled via a custom MATLAB script.

## Analysis of auditory evoked responses

Raw EEG data were transformed into a MATLAB compatible format (.mat) using the *tdtbin2-mat* MATLAB function (provided by TDT). Evoked responses were analysed within a time window set between -0.5 s and +5 s relative to sound stimulus onset. Each stimulus presentation is referred to hereafter as a trial. Trials that fell into an epoch that contained an artefact (as defined during the manual vigilance state scoring procedure) were excluded from further analysis. To aid computing efficiency, the signal was downsampled by a factor of 2 (from an

original sampling rate of 24414 Hz). Trials were then sorted into groups based on condition, stimulus frequency, level and vigilance state.

Due to marked inter-trial-variability in the EEG signal, conventional averaging across trials did not allow to define peaks and troughs of auditory evoked potentials (AEPs) reliably. To reduce the impact of noise on AEP detection, 20 bootstrapped means were drawn to serve as a representative signal sample for subsequent analysis, allowing for detection of peaks and troughs with minimal effect of noise while still reflecting the variability in the original dataset. Bootstrapped means were drawn from each group of trials (all trials for the same condition, vigilance state, stimulus and stimulus intensity), respectively. Peaks and troughs of evoked potentials were detected after smoothing each of those signals using a moving average of ~8 points (implemented by the MATLAB *smooth* function). Peaks and troughs were automatically detected in a time window after stimulus onset that was predefined using a custom written MATLAB script and the *findpeaks* function. Time windows for early (R1), mid (R2) and late (R3) response components were defined based on the average latency of peaks and troughs in the respective animal, as the shape of the evoked potential was not uniform across animals. R1, R2 and R3 could each be identified in Ferret 2, R1 and R2 in Ferret 3, and just one response component, R1, in Ferret 1. Note that the response windows for R1, R2 and R3 components of the response were defined for each animal individually, depending on the latency of the respective deflection relative to stimulus onset. Response magnitudes for each response component were then defined as the difference between the maximum and subsequent minimum of the signal in the response window for the respective response component. For each animal, magnitudes of all response components were included (pooled, means±SEM) in the analysis.

## Final recordings with Neuropixels probes in the auditory cortex

In two out of the three implanted cases (Ferret 2 and Ferret 3) and in one additional control, we performed terminal recordings under general anaesthesia in the left auditory cortex (contralateral to the noise overexposured ear) using Neuropixels probes.

General anaesthesia was induced and maintained using Domitor (0.022 mg/kg/h) and Narketan 10 (5 mg/kg/h) in 0.9% sodium chloride, pH 7.2–7.4 with 5% glucose delivered intravenously (3–5 ml/h) for the complete duration of the recordings (24–36 hours). After tracheal intubation, the animal was artificially ventilated. Atropine sulphate (0.06 mg/kg s.c.), doxapram hydrochloride (4 mg/kg, s.c.), dexamethasone (0.5 mg/kg s.c.) and viscotears were administer every 6 hours to minimize pulmonary secretion, reduce respiratory depression, prevent cerebral oedema, and protect the eyes, respectively. Perioperative analgesia and antibiotics were also administered during the surgery as mentioned for the previous surgery, and depth of anaesthesia, respiratory rate, ECG, and end-tidal $CO_2$ were monitored. Rectal temperature was maintaining at 38°C with the help of a homeothermic electric blanket and a forced air-warming system.

After removing the chronic implant in the NOE animals, a craniotomy (4x4 mm$^2$) was performed over the ectosylvian gyrus to reveal the auditory cortex. The animal was placed in an anechoic chamber and the dura mater was opened to visualize the primary auditory cortex.

Neural recordings were performed under anaesthesia (mixture of Domitor (medetomidine hydrochloride, 0.022 mg/kg, Orion Pharma; and Ketaset (ketamine hydrochloride, 5 mg/kg, Fort Dodge Animal Health) using Neuropixels 3A probes (single shank with 384 channels) and streamed to disk with SpikeGLX (https://billkarsh.github.io/SpikeGLX/) (sampling rate 25 kHz and filtered between 300 and 3000Hz). Spike clusters were identified using Kilosort2 spike sorting algorithm [56], curated by Phy (https://github.com/kwikteam/phy) and further analyzed using custom-written scripts in Matlab (R2023a). We analyzed the spontaneous firing

behaviour of clusters that were responsive to broadband noise (duration 300ms, intensity 60–90 dB SPL), as defined by spike counts in the first 100 ms window (1ms bins) following the stimulus presentation being significantly different from a window of the same duration preceding the stimulus (paired t-tests, P<0.05).

## Figures and illustrations

Figures and illustration were produced by using MATLAB and MS PowerPoint.

## Results

### Ferrets develop long–term behavioural impairments indicative of tinnitus after noise overexposure

To establish an animal model of induced, persistent tinnitus, adult female ferrets (n = 7) were tested in two operant paradigms sensitive to tinnitus (silent gap detection and silence detection) before (baseline, BL), starting one week after the unilateral noise exposure and, in a subgroup of ferrets (n = 3) also retested six months after the unilateral noise exposure (Fig 1B–1D). The protocol for noise overexposure (2 hours of one octave narrowband noise centred at 8kHz at 98 dB SPL) was similar to protocols used for tinnitus induction in other animal models [50,57,58]. Noise overexposure triggers chronic tinnitus in human and animal models whereas salicylate models evoke reversible tinnitus [18,59,60]. Our experimental design was combined with regular assessment of auditory brainstem responses (ABRs, see Methods).

The tinnitus index was calculated as the sum of three metrics obtained by comparing the values before and after the noise overexposure, with positive values indicating changes in line with tinnitus development (see Methods).

$$TI = M(cont) + M(thresh) + M(silence) \qquad \text{Eq(1)}$$

The metrics for operant gap detection performance refer to the change in the ability of an animal to detect the continuous sound (no gap) across all tested stimulus frequencies and to the change in gap-detection threshold across all stimuli (Eq(2) and Eq(3) in the Methods). The metric for operant silence detection refers to the change in silence detection ability (percent correct score in silence trials) after noise overexposure (Eq(4) in Methods).

The primary operant paradigm was operant silent gap detection, which assessed the animals' ability to detect short silent gaps in auditory stimuli [53] (Fig 1B and 1C). Four animals were implanted for chronic recordings but only three were used for long term assessment. No behavioural differences were found between implanted (IM) and non-implanted (NIM) animals in baseline assessments and therefore behavioural data from the seven ferrets were analysed together (operant gap detection: BL threshold (IM) = 3.36±0.81 ms; BL threshold (NIM) = 4.3±2.0 ms. Effect of group $F_{(1,32)}$ = 0.04, p = 0.85 (n.s.)).

In line with previous work on silent gap detection in ferrets [53], the animals' performance declined as gap length was reduced (effect of gap length, GLMM, $F_{(7,6752)}$ = 4489.37, p<0.001), an effect visible throughout the experiment (S1A Fig). After noise overexposure (NOE), silent gap detection ability was impaired, as indicated by the progressively lower percentage correct scores achieved at most gap lengths (S2A Fig) and by progressively increasing silent gap detection thresholds measured across all stimulus types (silent gap length at threshold, Baseline, BL: 4.0±1.78ms; One week: 5.77±2.68ms; Six months: 10.45±3.5ms, means±SD; GLMM, BL vs One week, β = 1.69,$t_{(75)}$ = 2.41, p = 0.018; BL vs Six months, β = 5.95,$t_{(75)}$ = 11.68, p<0.001; One week vs Six months β = 5.46, $t_{(75)}$ = 2.24, p = 0.029. Figs 1E and S2A).

The initial impairment observed (GLMM, effect of condition, $F_{(2,941)}$ = 32.31, p<0.001) was mostly due to reduced silent gap detection ability for 8 kHz narrowband noise burst (NBN) stimuli (probability of correct response, BL vs 1 week, 0.77±0.26 vs 0.7±0.27, β = -0.1, $t_{(943)}$ = -8.03, p<0.001), the same as the NOE sound, whereas performance remained stable for other stimuli (p>0.1) (S1B Fig). Six months after NOE, the impairment was also present at neighbouring frequencies (4 and 16 kHz NBN) (4 kHz BL vs 6 months, 0.75±0.27 vs 0.64±0.35, β = -0.04, $t_{(1319)}$ = -3.35, p<0.001; 16 kHz BL vs 6 months, 0.76±0.26 vs 0.6±0.31, β = -0.14,$t_{(1359)}$ = -11.79, p<0.001. S1B Fig). The stimulus that differed most from the NOE sound in terms of frequency composition (1 kHz NBN) was the least affected across the course of the experiment (p>0.1). Accordingly, the stimulus that comprises a broad range of frequencies (BBN) was also less affected.

While silent gap detection ability progressively worsened after NOE, detection ability for no-gap stimuli improved over time (S2B Fig). Specifically, ferrets showed statistically significant lower FA rates starting 1 week following NOE for 1, 4 and 16 kHz NBN (1 kHz, BL vs One week, 0.36±0.15 vs 0.27±0.15, β = -0.1,$t_{(162)}$ = -2.66, p = 0.01; 4 kHz, BL vs One week, 0.34 ±0.15 vs 0.24±0.12, β = -0.12,$t_{(162)}$ = -2.84 p = 0.01; 16 kHz, BL vs One week, 0.34±0.16 vs 0.26 ±0.14, β = - 0.08,$t_{(169)}$ = -8.75, p<0.001) stimuli, but not for 8 kHz NBN, the same stimulus as the NOE sound, and for BBN (p>0.1) (S1C Fig). This suggests that animals developed a temporary impairment in detecting both gap and no-gap sounds consisting of 8 kHz NBN, whereas in frequency ranges adjacent to the NOE stimulus, they were more likely to respond to stimuli as if they were no-gap sounds. In the longer term, six months after NOE, the FA rates were significantly lower for all stimuli, including 8 kHz NBN (p<0.001, S1C Fig) whereas the proportion of correct responses for gap stimuli was further impaired (S1B Fig). This improvement in FA rate over time, together with the gap detection impairment, suggest a tendency of the animals towards interpreting sounds with silent gaps as continuous sounds across all tested frequencies and could be an indication for tinnitus development.

To assess whether the impairment in silent gap detection ability after NOE could be attributed to diminished temporal resolution in auditory processing, the ferrets were tested in a silence detection paradigm [55]. They were tested in the same operant arena as for silent gap detection but had to discriminate NBN bursts of varying frequency compositions from 'silence', i.e. trials without any presented auditory stimulus, and from amplitude modulated (AM) BBN (see Methods & Fig 1D). As with operant silent gap detection, no behavioural differences were found between implanted (IM) and non–implanted (NIM) animals on silence detection and therefore behavioural data from the seven ferrets were analysed together (Proportion of correct responses: Amplitude modulated (AM) sound, IM vs NIM 0.99±0.01 vs 0.98 ±0.01, $F_{(1,6)}$ = 4.03, p = 0.09. Narrow band noise (NBN), IM vs NIM 0.99±0.01 vs 0.96±0.05, $F_{(1,6)}$ = 1.76, p = 0.23. Silence: IM vs NIM 0.92±0.06 vs 0.91±0.08, $F_{(1,6)}$ = 0.05, p = 0.84).

Animals were able to detect AM and NBN stimuli but less able to identify silence (Figs 1F and S1D). Animals with tinnitus might be expected to confuse NBN stimuli with an internally generated percept and therefore show a bias towards responding during silence trials as if NBN stimuli had been presented (Fig 1D).

At both timepoints following NOE, NBN detection performance on this task was similar to baseline (NBN, BL vs One week, 0.97±0.04 vs 0.97±0.02, β = 0.01,$t_{(16)}$ = 0.32,p = 0.76, BL vs Six months, 0.97±0.04 vs 0.91±0.1, β = -0.03, $t_{(16)}$ = -2.19, p = 0.05) (Fig 1F), whereas a significant decrease in AM detection performance was evident six months later (AM: $F_{(2,14)}$ = 6.57, p = 0.01, BL vs One week, 0.99±0.01 vs 0.99±0.01, β = -0.001, $t_{(16)}$ = -0.19, p = 0.86; BL vs Six months, 0.99±0.01 vs 0.98±0.01, β = 0.004, $t_{(16)}$ = 2.39, p = 0.03). The animals also achieved lower scores for silence trials following NOE, but this difference was not significant (Silence: BL vs One week, 0.91±0.07 vs 0.8±0.83, β = -0.17, $t_{(16)}$ = -2.15, p = 0.05; BL vs Six months, 0.91

±0.07 vs 0.83±0.06, β = -0.06, t$_{(16)}$ = -0.87, p = 0.4) (Fig 1F). However, note the relatively high variability in the silence trials, also in line with previous studies using the same paradigm [55,61].

Response times for both correct and incorrect trials, although initially unchanged in the trials in which auditory stimuli were presented (AM: BL vs One week, 1.38±0.12s vs 1.5±0.21s, β = 0.03, t$_{(16)}$ = 2.2,p = 0.05; NBN: BL vs One week, 1.43±0.1s vs 1.49±0.16s, β = 0.02, t$_{(16)}$ = 1.21, p = 0.25), became significantly longer six months after NOE (AM: BL vs Six months, 1.38 ±0.12s vs 1.77±0.26, β = 0.09, t$_{(16)}$ = 6.61,p<0.001; BL vs Six months, 1.43±0.1s vs 1.94±0.47s, β = 0.1, t$_{(16)}$ = 2.23, p = 0.04) (S2C and S2D Fig). In silence trials, longer response times were initially present in 'correct' trials (BL vs One week, 3.16±0.54s vs 3.95±0.95s, β = 0.1, t(16) = 3.08, p = 0.01) (S2C Fig), but had returned to baseline levels six months after NOE (p = 0.1) and were unchanged in 'incorrect' trials (p>0.1) (S2D Fig). Notably, response times in silence trials six months after NOE were nearly identical between animals, which may indicate a stereotyped response across animals.

The increased response times after NOE in trials where a sound stimulus was present may, in combination with the small bias of the animals to respond as if a sound was presented in silence trials, indicate increased uncertainty in the animal's perception about whether a sound was present or not. It seems unlikely that animals had difficulty discriminating between the sound stimuli (AM and NBN) as performance was not affected for NBN trials and only minimally so for AM trials (proportion of correct responses, Fig 1F). Therefore, the long–term effects seen in this paradigm are consistent with the perception of a phantom sound.

To assess the integrity of the auditory brainstem over time, ABRs were measured in all animals one week after NOE (see Methods; S1E Fig). In agreement with previous studies in ferrets [62–64], ABRs presented high variability across individual animals, which, compared to rodents, is likely due to the increased thickness of the skull. However, ABRs showed robust and reliable peaks and troughs with highest sensitivity between 8 and 16 kHz (Fig 1G), corresponding to the highest sensitivity in the ferret audiogram [65].

Following NOE, ABR thresholds significantly increased for stimuli with 8 kHz centre frequencies, the NBN NOE stimulus (BL vs One week, β = 15.71, t$_{(16)}$ = 7.24, p<0.05), and above (16 kHz, (BL vs One week, β = 15.71, t$_{(16)}$ = 2.17, p<0.05) (Fig 1G, Table 1). In the late assessment (Six months), thresholds were significantly elevated across all tested frequencies (p<0.05), except for 1 kHz NBN, suggesting a long–term degradation of auditory function.

The temporally frequency-specific ABR impairment following NOE suggests that behavioural changes affecting wider frequency ranges surrounding the NOE stimulus (such as reduced false alarm rate in operant silent gap detection) cannot be entirely ascribed to hearing loss. Therefore, the animals likely developed an initial NOE–frequency specific hearing impairment and eventually hearing loss for the NOE stimulus and adjacent frequencies, but also tinnitus affecting a wider range of frequencies in auditory silent gap detection.

The seven animals presented specific behavioural and hearing impairments without any noticeable general change in demeanour or wellbeing following noise overexposure. No changes in body weight, bowel habits, fur aspect, or social interaction were observed that could indicate potential noise overexposure related distress (see Methods). The surgical procedure undergone by the three ferrets is unlikely to have contributed to the observed hearing impairment since previous experiments [46] indicate no effects of comparable cranial surgery on auditory cortical responses in adult ferrets of the same age as in the current study when tested 30 months later.

**Table 1. ABR thresholds across time (in dB SPL).** Each table shows the threshold for a given stimulus (1, 2, 4, 8, 16 kHz centred NBN and BBN). Each row depicts data from one animal. Note that a subgroup of ferrets (n = 3) only was assessed at six months after NOE.

| 1 kHz | | | 2 kHz | | | 4 kHz | | |
|---|---|---|---|---|---|---|---|---|
| **BL** | **1 week** | **6 months** | **BL** | **1 week** | **6 months** | **BL** | **1 week** | **6 months** |
| **80** | 70 | 90 | 60 | 50 | 90 | 60 | 50 | 90 |
| **70** | 60 | 90 | 50 | 60 | 90 | 60 | 50 | 90 |
| **80** | 80 | 70 | 70 | 60 | 80 | 60 | 60 | 60 |
| **80** | 50 | | 80 | 40 | | 50 | 50 | |
| **90** | 90 | | 70 | 70 | | 50 | 70 | |
| **70** | 90 | | 70 | 90 | | 50 | 80 | |
| **90** | 70 | | 60 | 70 | | 40 | 60 | |
| 8 kHz | | | **16 kHz** | | | **BBN** | | |
| **BL** | **1 week** | **6 months** | **BL** | **1 week** | **6 months** | **BL** | **1 week** | **6 months** |
| **50** | 50 | 90 | 40 | 90 | 90 | 70 | 60 | 70 |
| **40** | 50 | 90 | 40 | 40 | 90 | 50 | 70 | 90 |
| **40** | 50 | 60 | 40 | 40 | 40 | 80 | 70 | 90 |
| **50** | 40 | | 40 | 40 | | 70 | 70 | |
| **50** | 80 | | 40 | 50 | | 70 | 90 | |
| **40** | 90 | | 40 | 80 | | 70 | 90 | |
| **40** | 60 | | 40 | 50 | | 60 | 80 | |

## Sleep-wake architecture following noise overexposure

To assess changes in sleep–wake distribution in parallel with the emergence of tinnitus after NOE (Fig 2), three adult female ferrets were implanted chronically with EEG electrodes (frontal and occipital derivation, following standard configuration [36,42,49] (see Methods, Fig 2C), and brain activity was continuously recorded in freely behaving animals for periods of 48 h under baseline conditions, and, as before, one week and six months following NOE (Fig 2A and 2B). We identified four different vigilance states: wakefulness, NREM sleep, REM sleep and a previously described secondary REM sleep, REM2 [42] (Fig 2D and 2E).

The animals spent most of the time (75.9±4.86%) sleeping in undisturbed baseline conditions (71.5%, 85.6%, and 70.6%, Fig 2A–2C), similar to previously reported sleep durations in ferrets (70.34±1.69%, [42]). Durations in vigilance state were similar across days in each ferret (Fig 2F) and variability between animals closely matched previous reports [42]. Sleep was dominated by non–rapid eye movement (NREM) sleep in all individual animals (Fig 2F). The remaining sleep time was predominantly spent in REM sleep, except in one animal, which spent more time in REM2 than in REM (Fig 2F). In line with previous experiments [42], animals did not manifest strong diurnality and slept both during the light and dark periods (Fig 2G) although the animals' activity typically increased after light onset for at least 2 hours (2–7 hours across animals; Fig 2G).

Following NOE, all animals developed behavioural signs of tinnitus, as measured by the tinnitus index, TI (see Methods) (Figs 3A and S3). In two animals, tinnitus was most pronounced six months after NOE. In addition, animals developed progressive hearing impairment, which was most pronounced in the animal with the least evidence for tinnitus (Ferret 3, Figs 3A and S3).

The sleep pattern changed in all animals following NOE, although this differed between individual ferrets: in both animals with strong signs of tinnitus and weak hearing impairment, sleep became disturbed after NOE but at different time points (Ferret 1 and 2, Fig 3C), while in the animal with weak indication of tinnitus but pronounced hearing impairment (Ferret 3,

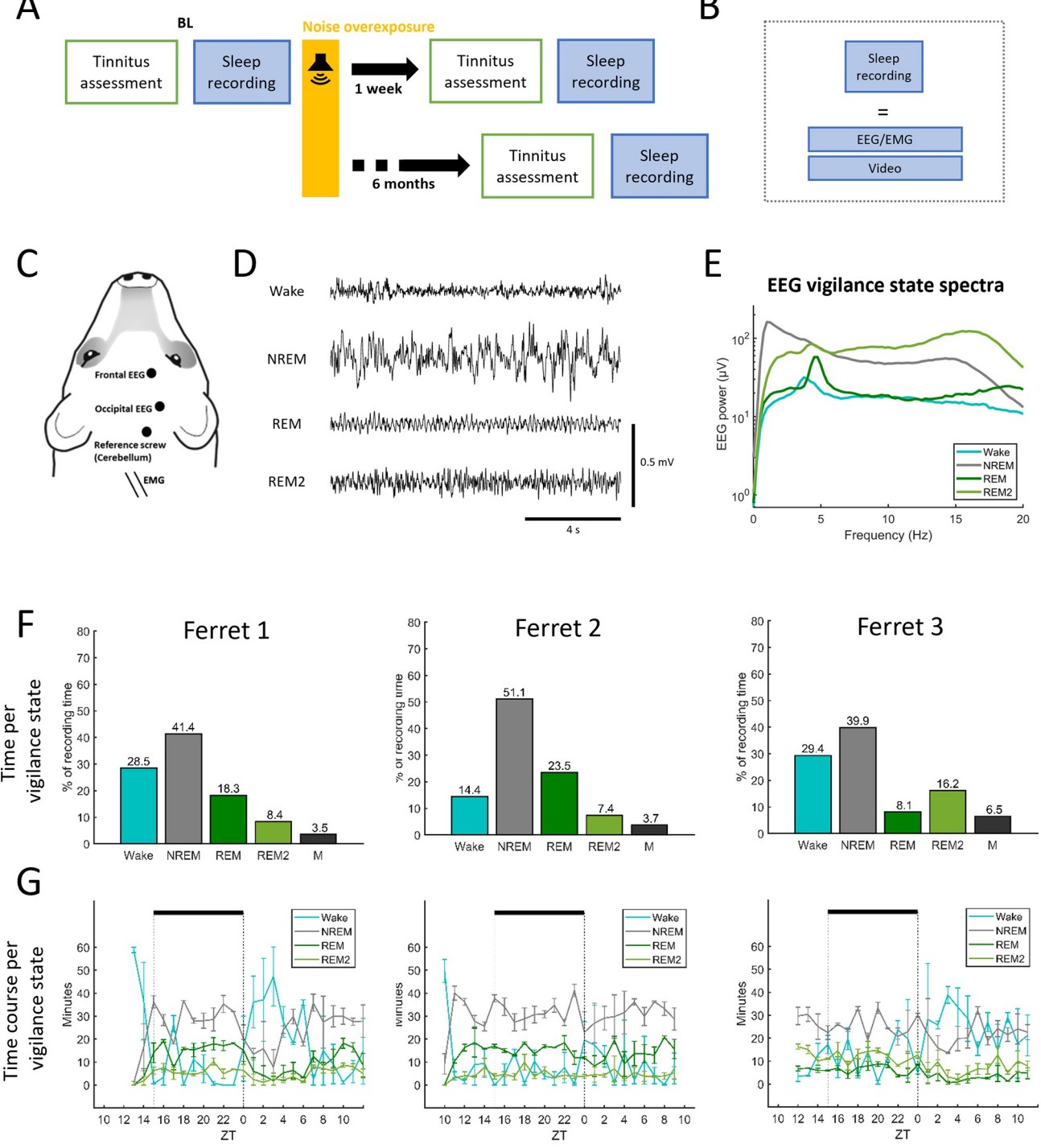

**Fig 2. Chronic recordings during sleep and wakefulness.** (**A**) Experimental timeline for implanted animals. Three ferrets were assessed for tinnitus, hearing loss, EEG brain activity and sleep-wake behaviour before and after noise overexposure (NOE). Assessments of all these metrics took place once under baseline conditions (BL) and on two occasions after NOE, the first assessment (One week) commencing one week following NOE and the second starting six months following NOE. In addition to the 'tinnitus assessment', (Fig 1), brain activity was recorded (based on frontal and occipital EEG) in the freely behaving animals for approximately 48 hours in each condition. (**B**) Sleep recording consists of EEG/EMG and continuous video recording. (**C**) Depiction of the ferret head showing the position of the implanted screw electrodes for EEG recordings (frontal and occipital), the ground reference electrode implanted over the cerebellum and the EMG wire electrodes on a schematic ferret head. (**D**) Example EEG traces during wakefulness (Wake), non-rapid eye movement sleep (NREM), rapid eye movement sleep (REM) and REM2 sleep (REM2). EEG signals displayed in this panel are band-pass filtered (0.5–30 Hz) and were obtained under baseline conditions (before NOE). (**E**) EEG vigilance state spectra in the ferret (example based on Ferret 3).

Data are means across EEG spectra of the frontal derivation calculated for two consecutive 24 hour recordings, based on 0.25 Hz frequency bins. Shading depicts the standard error. Vigilance states (Wake, NREM, REM and REM2) are colour-coded (see inset figure legend). (**F**) Amount of wakefulness and sleep under baseline conditions for each ferret. M, movement artefacts within sleep episodes. (**G**) Time course of wakefulness and sleep under baseline conditions for each ferret. Zeitgeber time (ZT) represent the start of light period.

Fig 3A and 3B) sleep became progressively more stable after NOE, with fewer occurrences of wake episodes during sleep (Fig 3C).

To evaluate the individual impact of tinnitus and hearing loss on sleep, each animal was analysed independently with respect to changes in sleep–wake architecture before and after NOE.

## Ferret 1: Progressive tinnitus, mildly raised auditory thresholds and long–term sleep stability

This animal showed marked signs of tinnitus in the first assessment following NOE (TI 0.5, One week) and a progressive increase towards a TI of 0.9 six months later (Figs 3A and S3). In addition, it showed progressive mild hearing impairment following NOE with threshold elevations of 10 dB (One week) and 11.7 dB (Six months) (Figs 3A and S3) and a reduction of total ABR magnitude by 4.1% (One week) and 17.7% (Six months) (S3 Fig).

Sleep amount increased transiently following NOE (Figs 3B and S4), in combination with elevated NREM EEG slow–wave activity (SWA, EEG power density between 0.5–4 Hz) (S4 and S5 Figs). It is possible that a change in sensory experience following NOE results in compensatory plasticity that is associated with increased sleep need, although increased sleep disruption (Fig 3C) may have contributed to the elevated sleep need in this animal. In the longer term the animal showed a reduction in sleep amount (Wake BL vs Six months, 29.1% vs 34.2% of recording time, Fig 5, grey bars) but also showed reduced sleep disruption (Fig 3C). Note that despite the strong behavioural indication for tinnitus in this animal six months following NOE, sleep amount was largely unchanged as compared to baseline conditions.

## Ferret 2: Stable tinnitus, progressive changes in brainstem activity and long–term disturbed sleep

Ferret 2 showed evidence for stable tinnitus, which emerged soon after NOE (Fig 3A) and was initially (One week) of similar intensity to Ferret 1. In parallel, the animal initially showed evidence of increased sleep propensity, with less disrupted sleep than before NOE (70 wake episodes in baseline versus 49 in One week). However, six months following NOE, sleep disruption was markedly increased with almost double the number of wake episodes (136 vs 70 wake episodes in BL, Fig 3C), most of which were rather brief (<5 minutes, Fig 3B).

Hearing impairment following NOE was reflected in a progressive ABR threshold elevation (One week: 13.3 dB, Six months: 25 dB, Figs 3A and S3), suggesting an impairment in auditory sensitivity. Furthermore, total ABR magnitude was temporarily reduced in One week (-31.2%), but partially recovered subsequently (Six months: -7.7%, Figs 3A and S3). This later recovery may indicate a long–term compensation following reduced peripheral input through central or peripheral gain elevation.

The initial changes in brainstem evoked activity after NOE were paralleled by a temporary reduction in time spent awake (BL vs One week: 15.1% vs 12.9%, Fig 5, grey bars) and reduced sleep disruption (Fig 3B). This could be due to temporarily increased sleep pressure following NOE, as also indicated by significantly increased EEG slow–wave activity during NREM sleep (BL vs One week, Two–way ANOVA, Tukey's multiple comparisons, p<0.05, S4 and S5 Figs) and during REM sleep (BL vs One week, p<0.05, S4 Fig). As in Ferret 1, it is possible that

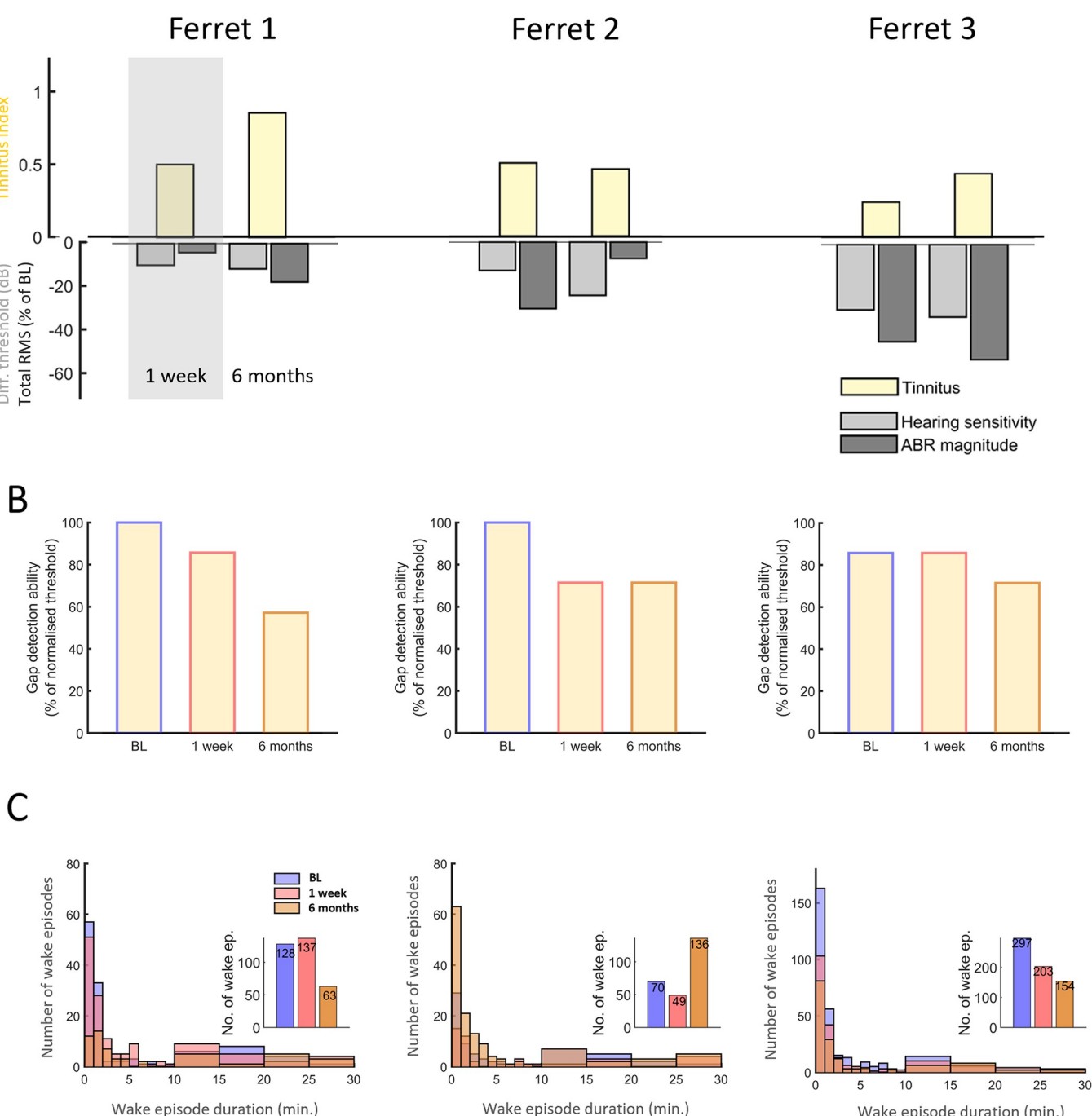

**Fig 3. Tinnitus development, hearing loss and sleep disruptions assessed over time. (A)** Tinnitus and hearing loss development over time in the three implanted ferrets (Ferrets 1–3). Yellow bars depict behavioural evidence for tinnitus (using the tinnitus index, TI, for each animal (based on operant gap and silence detection; see Methods for details)). Light grey bars show hearing loss based on the differences from baseline in ABR thresholds while dark grey bars show the change in ABR total RMS magnitude, respectively. The bars are a depiction of the metrics for TI and ABR changes displayed in S3 Fig. The size of each bar indicates the change relative to BL of the corresponding measure at 1 week and 6 months. Each panel represents a different ferret. Left to right: Ferret 1, Ferret 2, Ferret 3. **(B)** Impairment in gap detection ability over time based on thresholds. Displayed values correspond to the m(thresh) metric used to calculate the tinnitus index (TI, see panel **A**) for each individual case. Values are based on gap detection thresholds fitted on a continuous scale ranging from 0–100%. 100% corresponds to a gap detection threshold of 3ms, and the remaining gap lengths (5, 10, 20, 50, 270ms) correspond to the evenly spaced

decrements in threshold (see methods for details). Note that an increase in gap detection threshold over time is represented by a decrease in the values presented here. Blue framed bars are Baseline test (BL); red and orange framed bars correspond respectively to measurements conducted at 1 week and 6 months post noise exposure (**C**) Number of wake episodes during ~ 48 hours of baseline recording. Large panels are histograms depicting the number of wake episodes separated by episode duration and the insets are the total number of wake episodes for baseline (BL), one week and 6 months. The y-axis and the number displayed in the bars in the inset panels depict the number of wake episodes. Note the difference in y-axis scale for ferret 3. Panels left to right: Ferret 1, Ferret 2, Ferret 3. Note that in Ferret 2 the number of wake episodes of nearly all durations increase six months following NOE and overlap the other bars (Baseline and one week post NOE) in the panel. Experimental timepoints are depicted in blue (BL), red (1 week), orange (6 months).

compensatory plasticity after NOE led to increased sleep need reflected in SWA elevation during sleep. The animal's sleep subsequently became markedly more disrupted, with nearly twice the amount of wake episodes (Six months, Fig 3B) and with overall less sleep than in the baseline condition (Wake BL vs Six months: 15.1% vs 24.0% of recording time, Fig 5). This suggests that the animal may have become more sensitive to external stimuli (hyperacusis) or to internal triggers for arousal, such as tinnitus.

## Ferret 3: Mild tinnitus with progressive, pronounced hearing loss and progressive sleep stability

Ferret 3 showed the most pronounced hearing impairment and the least indications of tinnitus. Even though the TI increased over time, it was generally low (One week, TI = 0.2, Six months, TI = 0.4, Figs 3A and S3). ABR thresholds were markedly elevated following NOE (One week: by 30 dB, Six months: by 33.3 dB, Figs 3A and S3) and a progressively reduced total ABR magnitude (One week: -44.7%, Six months: -52.9%, S3 Fig) further indicated the presence of more severe hearing loss in this animal.

There were no signs of increased sleep disruption following NOE (based on the number of wake episodes, Fig 3B). To the contrary, in parallel to progressively impaired hearing, sleep became progressively less disrupted (Fig 3B) and the amount of time the animal spent asleep increased (Wake amount in BL 28.2%; One week 20.3%, Six months 23.1%, Fig 5). The decrease in sleep disruption following NOE may be linked to an elevation of the auditory arousal threshold due to hearing loss.

Different from the other animals, sleep in this animal was characterised by lower SWA following NOE than during BL, possibly suggesting more superficial sleep (Figs 5, S4 and S5). The increased sleep amount is unlikely to be a compensatory response to reduced sleep intensity. Instead, this was potentially a consequence of the animal's ability to maintain consolidated sleep for longer, accumulate a large amount of sleep overall and therefore experience less homeostatic sleep pressure, which is determined predominantly by the time spent awake.

In summary, while all three animals showed a progressive increase in ABR thresholds after NOE, the magnitude of this impairment differed across individuals, as did the emergence of behavioural signs of tinnitus and changes in sleep–wake architecture. Both animals with strong behavioural signs of tinnitus (Ferrets 1 and 2) showed initially higher sleep need after the noise trauma. In the longer term, sleep was disrupted to varying degrees. In the animal with severe hearing impairment and little evidence for tinnitus (Ferret 3), sleep maintenance (based on the number of sleep episodes) improved following noise overexposure.

**Increased evoked activity in tinnitus is modulated during sleep.** To assess whether changes in cortical excitability or responsiveness correlate with tinnitus and might underlie the observed differences in sleep pattern after NOE, auditory evoked activity was evaluated across all vigilance states using free–field sound presentation (see Methods). Briefly, after collecting undisturbed EEG recordings for 24 h, auditory stimuli were presented via a free–field speaker during the subsequent 24 h (Fig 4A). Sounds were one octave narrow band stimuli with centre frequencies of 1, 4, 8 and 16 kHz (820 ms duration, central gap 38 ms, see details in

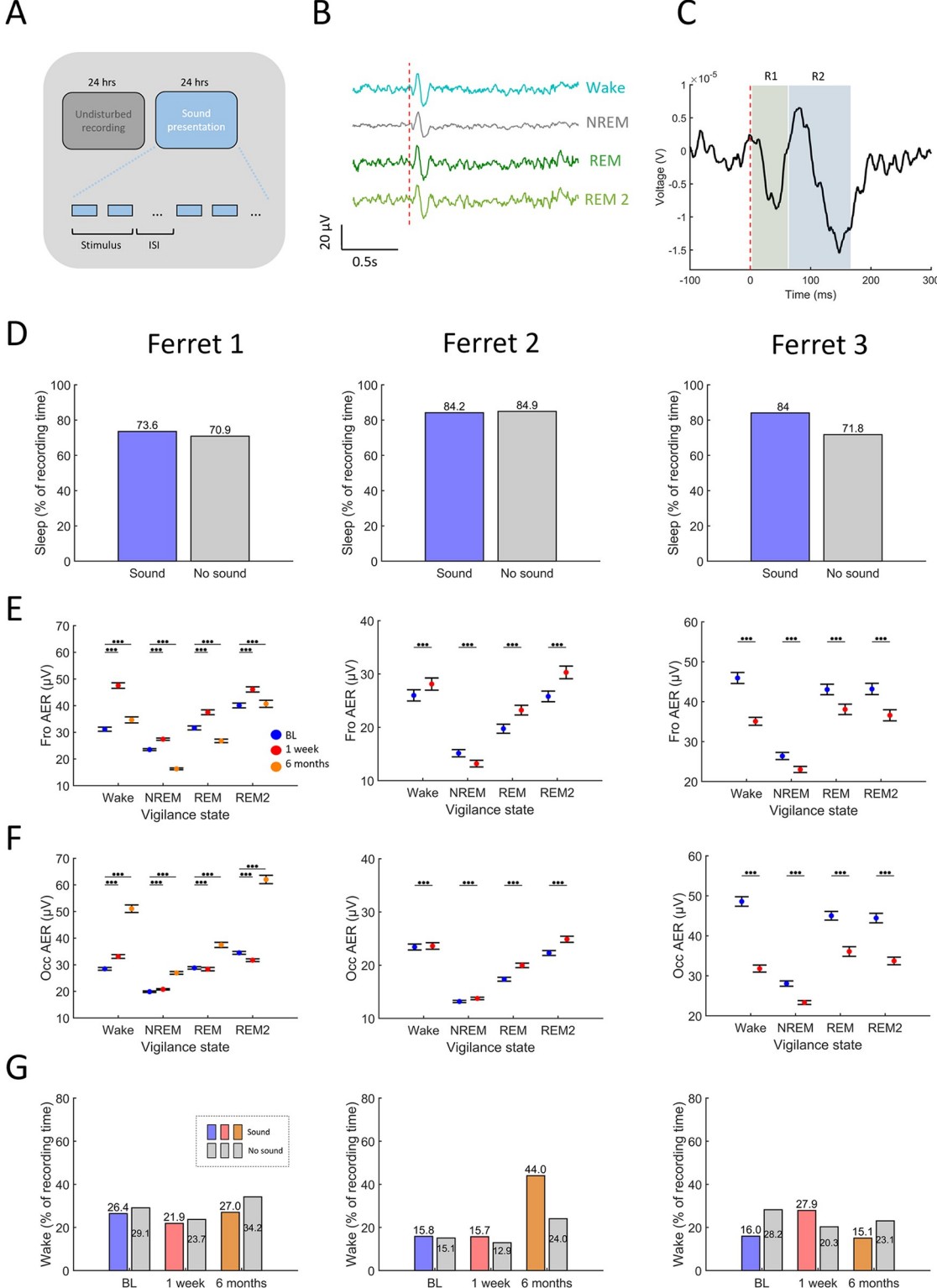

**Fig 4. Sound evoked cortical activity across vigilance states before and after NOE.** (**A**) Experimental paradigm. Sounds were presented through a single loudspeaker located at the top of the enclosure over a period of 24 hours subsequent to 24 hours of undisturbed recordings. Sounds were one octave narrow band noise (NBN) bursts centred at 1, 4, 8, and 16 kHz, at a stimulus level of 40, 50, 60, and 65 dB SPL, with a duration of 820 ms that included a silent gap of 38 ms. Stimuli were randomly presented with an interstimulus intervals of 10–42 seconds for a total number of 200 presentations per stimulus-level combination. (**B**) Average EEG

evoked responses during Wake, NREM, REM and REM2 sleep during the baseline condition. Signals are averages for one animal (Ferret 3) during the baseline condition. (**C**) Definition of evoked response magnitude. Evoked responses recorded from each animal were partitioned into components defined by the number of peaks and troughs in the signal. Evoked response magnitude was defined as the difference between the peak and subsequent trough for each response component (R1 and R2 in this example, see S6 Fig for Ferrets 1–3). (**D**) Sleep durations (as percent of recording time) in undisturbed conditions and with sound presentation measured in basal conditions before NOE. Each panel depicts one ferret. Panels left to right: Ferret 1, Ferret 2, Ferret 3. (**E**) Frontal EEG auditory evoked response (AER) across vigilance states before and after NOE (colour coded). Data are averages across response components, sound level and stimulus type based on bootstrap means ± standard errors (See methods). (**F**) Depiction as in **E** but for occipital EEG configuration. Asterisks for panels **E-F** represent statistical significance p values *<0.05, **<0.01, ***<0.001 (GLMM). Each panel depicts one ferret. Panels left to right: Ferret 1, Ferret 2, Ferret 3. Due to decreased signal quality in the six months post NOE assessment in Ferrets 2 and 3, AERs could not be quantitatively assessed for these timepoints. (**G**) Amount of wakefulness in baseline (BL), one week and 6 months post NOE recordings with sound presentation (coloured bars) and without sound presentation (grey bars), depicted as percent of recording time. Each panel depicts one ferret. Panels left to right: Ferret 1, Ferret 2, Ferret 3. Experimental timepoints are depicted in blue (BL), red (1 week), orange (6 months).

Methods). EEG auditory evoked responses (AERs) were obtained as shown previously in other animal models [66] during wakefulness and in all sleep states under baseline conditions (Figs 4B, 4C and S6–S9).

All animals spent the majority of time asleep while sounds were presented without marked differences according to whether sounds were presented or not (Fig 4D). This indicates that the presented sounds were of sufficient intensity to trigger evoked EEG responses but did not disrupt sleep.

There was a statistically significant interaction between the magnitude of the AER (pooled across all response components per animal, Figs 4C and S6, see Methods) and the vigilance state in both the frontal and the occipital EEG (*Frontal EEG*: Ferret1, $F_{(1,956)} = 1.874E+30$, p<0.001, Ferret2, $F_{(1,1915)} = 1.364E+31$, p<0.001, Ferret3, $F_{(1,1276)} = 1.423E+32$, p<0.001; *Occipital EEG*, Ferret1, $F_{(1,956)} = 8.092E+30$, p<0.001, Ferret2, $F_{(1,1916)} = 7.802E+24$, p<0.001, Ferret3, $F_{(1,1276)} = 4.892E+29$, p<0.001). AER magnitudes were lowest during NREM sleep and highest during Wake and REM2 sleep (Fig 4E and 4F, blue symbols), suggesting that NREM sleep may reduce sound–evoked activity. The modulation of AERs by the vigilance state was similar across sound intensities.

When tinnitus was more severe and hearing loss mild (Ferrets 1 and 2, Fig 3A), AERs during wakefulness increased after NOE (Fig 4E and 4F). This increase in responsiveness was attenuated during sleep, which could explain differences in sleep disturbance (number of wake episodes, Fig 3B) across ferrets with tinnitus. In Ferret 3, which showed pronounced and progressive hearing loss and less severe tinnitus, evoked activity was reduced across all vigilance states after NOE.

## Ferrets 1 and 2: Increased evoked activity in tinnitus is reduced during sleep

Ferret 1 showed a progressive increase in evoked activity (pooled across all response components, Figs 4C and S7), mostly in the occipital EEG derivation (Fig 4F), alongside behavioural evidence for tinnitus and little change in ABRs (Fig 3A). Elevation of evoked activity was less pronounced or absent during sleep (Figs 4E, 4F and S7).

The initial increase in evoked activity during wakefulness after NOE (One week) relative to BL was especially pronounced in the frontal EEG derivation (S7 Fig) and evident for all stimuli (p<0.001, GLMM). Nevertheless, evoked activity was still significantly elevated in the occipital EEG (p<0.001, GLMM), except for 4kHz NBN (S10 Fig, Ferret 1).

The increase in evoked activity after NOE depended on the vigilance state: it was most pronounced during wakefulness in both EEG derivations (Frontal EEG, BL vs One week, 31.17

±0.7 vs 47.53±1.0 µV; Occipital EEG, 28.47+−0.5 vs 33.11±0.7 µV means±SEM; Fig 4E and 4F). During NREM sleep, the increase was less pronounced (Frontal EEG, BL vs One week, 23.52±0.35 vs 27.43±0.43 µV; Occipital EEG, 19.88±0.27 vs 20.77±0.26 µV) or, as in the occipital EEG, even absent for REM and REM2 sleep (Fig 4E and 4F). Note that in the frontal EEG, while evoked activity remained lowest during NREM sleep even after NOE, it was still elevated compared to baseline.

In the late assessment, six months after NOE, the change in frontal evoked activity reversed and AERs for all stimuli approached baseline levels or below in the frontal derivation (S10 Fig), which was largely due to drastically reduced AERs during NREM and REM sleep (Frontal EEG: NREM, BL vs Six months, 23.52±0.35 vs 16.3±0.32 µV; REM, BL vs Six months, 31.64 ±0.74 vs 26.69±0.66 µV, Fig 4E). This could explain why the animal showed prolonged and less disrupted sleep six months after NOE. In the occipital derivation on the other hand, evoked activity increased further for all stimuli (S10, S11 and S14 Figs). Note that this pronounced increase in occipital evoked activity six months after NOE coincided with the largest behavioural tinnitus index of 0.9 among all animals (Figs 3A and S3) but was not associated with disrupted sleep.

In Ferret 2, AERs were increased relative to baseline in the first assessment after noise overexposure (One week) (p<0.001, GLMM) at both frontal and occipital EEG derivations (Figs 4E, 4F, S6 and S8) for all sound stimuli (S10 Fig). This supports the notion that the gain of auditory evoked responses increased following NOE.

As in Ferret 1, while the increase in evoked activity was evident in both EEG derivations, this was locally modulated across vigilance states: in the frontal EEG signal (measured across all components of the AER), NREM sleep was associated with a reduced evoked response after NOE relative to baseline (BL vs One week, 15.14±0.7 vs 13.19±0.62 µV, p<0.001, GLMM), whereas in the occipital EEG, the increase in evoked responses was evident in all vigilance states (p<0.001, GLMM, Fig 4E and 4F). Notably, the reduced frontal NREM AER was present for all frequency stimuli (S10 Fig). It is possible, therefore, that NREM sleep had a suppressing effect on frontal evoked activity after NOE.

Six months after NOE, Ferret 2 showed qualitative evidence for further increased auditory evoked activity in both frontal and occipital derivations (S12 and S14 Figs). Due to decreased signal quality in the Six months assessment, this could not be quantitatively verified. However, even without sound presentation, the animal showed an ~60% increase in the amount of time awake in the Six months assessment (BL vs Six months, 15.1% vs 24.0%, Fig 4G). With sound stimulation, this effect was amplified (time awake, BL vs Six months, 15.8% vs 44%, Fig 4G), while the amount of sleep was reduced (Fig 5), indicating that increased cortical responsiveness may have led to long–term sleep disturbance.

## Ferret 3: Generally decreased evoked activity after NOE and profound hearing loss

Ferret 3 showed reduced cortical evoked activity after NOE during all vigilance states (Fig 4E and 4F), in line with marked progressive hearing impairment (Fig 3A).

In the first assessment after NOE, auditory evoked activity (pooled across all response components; S6 and S9 Figs) was lower than in the baseline assessment (p<0.001, GLMM, Fig 4E and 4F). This was the case for all sound stimuli and for both the frontal and occipital EEG signals (p<0.001, GLMM, S10 Fig). The reduction in AERs was evident during all vigilance states for most stimuli. During REM2 and REM sleep, there were signs of elevated evoked activity for 8 and 16 kHz stimuli, respectively, but only in the frontal EEG signal (S14 Fig). Six months

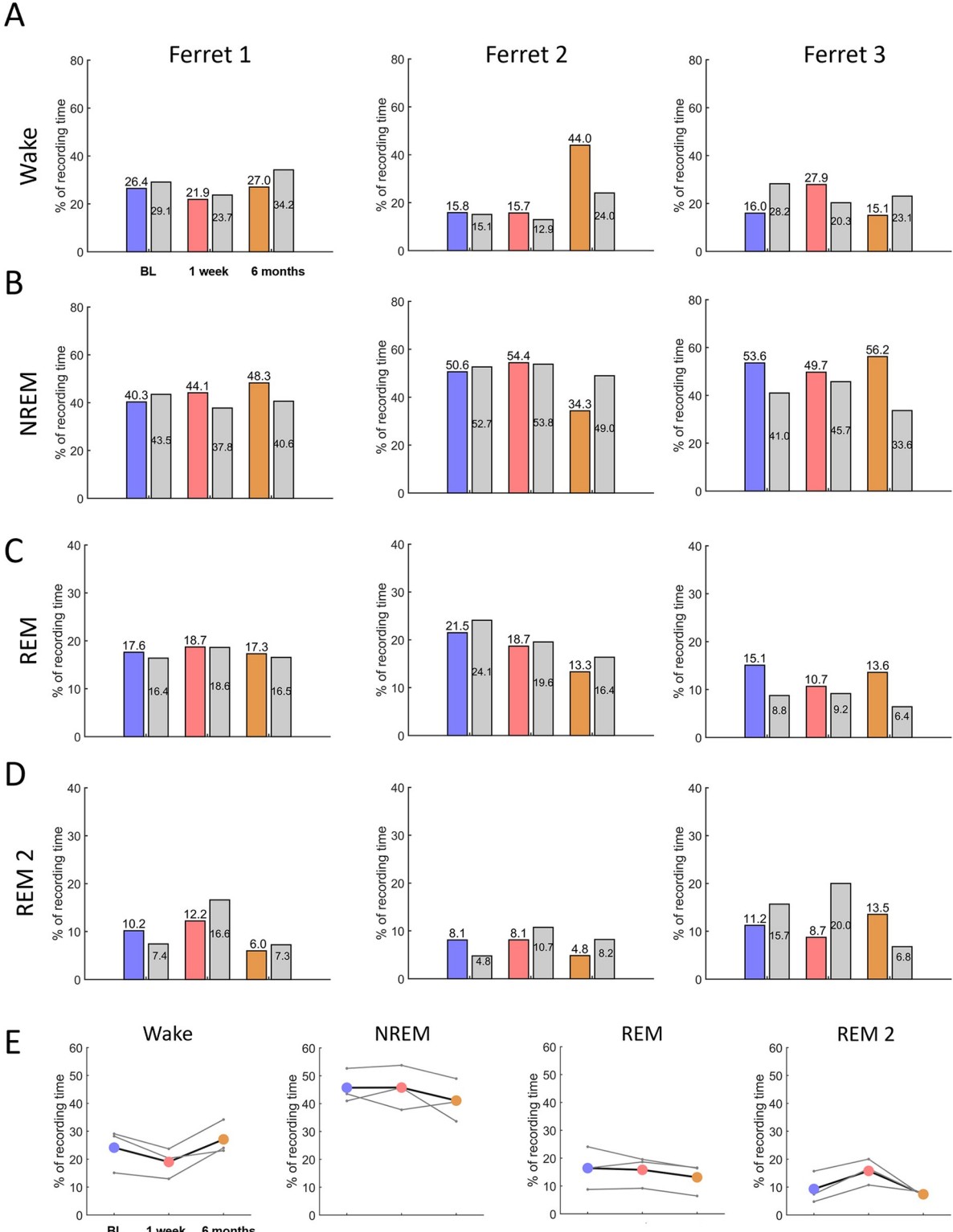

**Fig 5. Vigilance state durations with and without sound presentation. (A-D)** In each panel, colour bars show amount of time spent in wake, in NREM, in REM and in REM2 sleep, respectively, during 24 hrs of recording with sound presentation before noise overexposure (BL, baseline, blue), starting one week following NOE (red), and six months following NOE (orange). Grey bars show amount of time spent in the respective states without sound presentation. Numbers in the bar plots represent time spent in the corresponding state as a percentage of recording time. Each column of panels represents data from one animal (left to right: Ferret 1, Ferret 2, Ferret 3). (**E**) Average vigilance

state amounts during baseline (BL), one week and six months following NOE without sound presentation. Coloured markers and black lines depict averages and thin lines individual animal values.

after NOE, there were signs of signs of increased evoked activity in the occipital derivation (S13 and S14 Figs), but this could not be quantitatively analysed due to reduced signal quality.

While the progressive hearing loss in this animal after NOE in undistured condition (without sound stimulation) was associated with less sleep disruption (Fig 3C) and less time spent awake (Fig 4C), a similar trend over time was not apparent with sound stimulation (Fig 4G).

In summary, the two animals that showed greater behavioural signs of tinnitus also developed increased neural evoked activity after NOE although at different time points. The animal with less evidence for tinnitus but severe hearing loss showed reduced auditory evoked activity after NOE. When tinnitus was more severe and frontal evoked activity was elevated, sleep was disturbed, but not when occipital evoked activity was elevated. This might have important implications for the notion of frontal or brain–wide tinnitus representation playing a role in sleep disruption. In both animals with strong evidence for tinnitus, auditory evoked activity was lowest during sleep, suggesting a role for natural brain state dynamics in modulating tinnitus–related activity.

**Neuropixels recordings in the auditory cortex.** In two out of the three implanted cases (Ferret 2 and Ferret 3) and in one additional control, we performed final recordings in the left auditory cortex (contralateral to the noise overexposured ear). Acoustically-responsive cortical units recorded in NOE Ferrets 2 (median/interquartile 0.958/5.527) and 3 (md/iqt 2.713/5.764) exhibited a significantly greater median spontaneous activity than units recorded in the control ferret (md/iqt 0.275/1.975) (Kruskal-Wallis, $\chi^2_{(2,1073)}$ = 122.83; p<0.0001) (S15A Fig). The cortical units also had a broader distribution of firing rates after NOE than in the control animal (S15B Fig), indicating that the observed increase in median spontaneous firing rate was due to an overall increase with fewer units exhibiting a low spontaneous firing rate. Bursting activity was very low for all ferrets (Control: 0.025 Hz, Ferret 2: 0.012Hz and Ferret 3: 0.004 Hz) with only 12.6% (30/237 units) in Ferret 2, 5.4% (16/294 units) in Ferret 3 and 9.6% (52/543 units) in the control ferret producing at least one burst of spikes.

To gain an insight into the temporal precision of the cortical responses, we calculated for each unit the change in the interspike intervals (ISI) during their evoked activity (100 ms after stimulus presentation) compared to their spontaneous activity. Because we only analyzed acoustically-responsive units, the increased number of spikes would result in shorter ISI and therefore negative differences as was observed for all three ferrets, control and NOE (S15C Fig). Although we cannot fully untangle the contribution of the change in the excitability of the units to the reduction in the ISI, it is worth noting that such shortening was more pronounced in control units than after NOE. The observed differences in spontaneous activity between ferrets and less marked changes in the ISI are together suggestive of reduced temporal precision in the cortical activity in the NOE animals, which is consistent with the behavioural changes exhibited by those animals.

## Discussion

Tinnitus is thought to arise from aberrant spontaneous brain activity [1,15,18], yet the natural brain state, where activity is most dominated by spontaneous brain dynamics [29–31], has not been systematically investigated. Given the anatomical overlap of brain areas involved in both tinnitus and sleep [22,67–69], we hypothesized a bi-directional relationship, where tinnitus induces sleep alterations and sleep modulates tinnitus-related brain activity [32].

To address this hypothesis, we introduced a novel animal model—the ferret—to investigate chronic tinnitus, sleep, and their interactions. This model enabled us to track tinnitus development over a period of months following noise overexposure. By combining this with the assessment of sleep architecture and spatiotemporal brain activity, our measurements at two time points, one week and six months, following noise overexposure provide initial evidence that tinnitus emergence might coincide with emergence of sleep disruption. Furthermore, increased auditory evoked activity in tinnitus animals was reduced during sleep, suggesting a potent role for natural brain state dynamics in modulating aberrant brain activity associated with the effects of noise trauma.

We characterized the ferret as a novel animal model of tinnitus for longitudinal studies. Ferrets have a hearing range that overlaps that of humans [70], they can learn sophisticated behavioural tasks [53,71,72] and they have a longer lifespan [6–8 years, [73]] than rodents [41,74,75]. We demonstrated that noise overexposure in ferrets is not only associated with changes in behavioural performance that are indicative of tinnitus, but also with hearing impairment. Although unexpected because we used the same settings established in other animal models (Unilateral exposure for 1–2 hours of one octave narrowband noise) [45–47] we found ABR threshold shifts that were not transient. Interestingly, we show that the measures of the degree of tinnitus or its perception by the animal, as well as hearing impairment following noise overexposure are highly idiosyncratic, in line with variable effects of noise trauma seen in other animal models and the tinnitus heterogeneity that is characteristically found in humans (for example reviewed in [76,77], respectively). Therefore, while future studies with a larger number of implanted animals and a specific control group are necessary to investigate the relationship between sleep and tinnitus and identify consistencies, the detailed study of individual cases will remain important, as increasing the number of animals and averaging the data may dilute and potentially mask important individual profiles. At the same time, work with larger groups with only tinnitus, hearing loss alone, hearing loss and tinnitus, and specific *ad hoc* controls is necessary to investigate this new area in greater detail and identify consistencies.

We obtained chronic EEG recordings to investigate changes in evoked activity and sleep-wake pattern in parallel to emerging tinnitus and hearing impairment. Interestingly, REM2 sleep amount increased in the three ferrets followed noise overexposure (Week 1) regardless of their different initial magnitude of tinnitus, although only transiently for ferrets 1 and 3 (Fig 5E). This may suggest that brain activity characteristic for this state, in particular oscillations in the beta and possibly gamma range, may be a marker for consequences of NOE. In humans exhibiting residual inhibition, gamma activity is positively correlated with tinnitus [78]. Although not many sleep studies had been performed in humans with chronic subjective tinnitus, polysomnography studies revealed pathological sleep patterns, with tinnitus patients remaining longer in shallow sleep (stages 1 and 2) and spending less time in deep sleep (stage 3) and REM sleep [4,79,80]. Those differences were not always statistically significant [81] but were more evident in patients where tinnitus was highly modulated by sleep [39], suggesting that tinnitus heterogeneity is always a factor to consider, both in human and animal studies. Future investigation may explore whether changes in cortical activity reflected by the REM2 state in the ferret also reflect initial tinnitus activity during sleep or the initial brain response to the noise overexposure.

We found that the single ferret developing severe and progressively worse hearing impairment after noise overexposure also developed progressively stable, prolonged and lighter NREM sleep. Building on previous findings showing that cochlear lesions can reduce wakefulness and prolong sleep [82], our results suggest that hearing impairment may increase sleep maintenance and lead to fewer awake episodes, likely as a result of increased sensory

disconnection. This differed in cases where tinnitus accompanied hearing impairment. Animals displaying more severe tinnitus following noise overexposure developed reduced and more disrupted sleep. While these results do not demonstrate a causal relationship between tinnitus–related aberrant brain activity and impaired sleep, the parallel emergence of tinnitus and sleep disturbance over time is highly suggestive of such a connection. Indeed, the elevated auditory-evoked cortical activity in animals with signs of severe tinnitus supports this possibility and is in line with previous studies reporting increased activity and excitability along the auditory pathway in tinnitus cases [23,83–86]. Cochlear damage after noise overexposure can lead to a compensatory increase in neuronal excitability that restores evoked activity. This can occur through mechanisms such as homeostatic plasticity [21] or adaptive stochastic resonance [87], processes that can themselves drive tinnitus development [19,21,87]. Elevated cortical activity has been reported in human tinnitus sufferers [88,89] and in a range of animal models of induced tinnitus (chinchillas: [86], cats: [90], guinea pigs: [91,92], gerbils [93,94], rats: [95], and mice: [96]), including our own preliminary results (S15 Fig).

The increased evoked activity we observed in the first assessment after NOE could be explained by adaptive stochastic resonance, which operates on short timescales [19]. In contrast, sustained tinnitus six months after NOE, especially in the context of higher levels of hearing loss, is more likely based on plastic changes [97] relating to homeostatic plasticity and alterations in central gain [21].

Assessments of tinnitus in our study are based on positive operant behavioural tasks rather than reflexive behaviours like the acoustic startle reflex, which show habituation that is particularly rapid in carnivores and primates, including humans [98]. However, there are further important differences in tinnitus identification based on operant and reflexive paradigms: for example, reflexive paradigms, such as classical assessment of the gap-induced inhibition of the behavioural startle response (GPIAS, [99,100]) make it challenging to separate tinnitus from hearing loss, whereas operant paradigms can be more sensitive to differences between these conditions (e.g. [55]). On the other hand, operant paradigms have a learning component that may affect the results (reviewed in [101]). For this reason, we completed extensive baseline training in all operant paradigms before testing animals for tinnitus (see methods).

It is possible that animals displaying signs of tinnitus in our study also developed other noise–induced conditions, such as elevated sensitivity to environmental sounds or hyperacusis. Our animals also developed different degrees of hearing loss and an increase in both spontaneous and driven activity in auditory cortical neurons, as has been reported in ferrets presenting moderate hearing loss following injections of the ototoxic drug kanamycin [102]. Although hearing loss and hyperacusis would not account for all the behavioural deficits observed in this study, such as impaired silent gap detection, it cannot be ruled out that hyperacusis was an additional consequence of noise overexposure and contributed to sleep impairments. Indeed, hyperacusis has been suggested as common factor in both tinnitus and insomnia [103].

The main limitations of this cross-case study are the small number of animals with chronic implants, and the absence of specific *ad hoc* controls. In addition, the elevation of ABR thresholds six months after NOE could have been a confounding factor as the relationship between hearing loss and tinnitus has long been recognised (e.g. [104], reviewed in [105]), even when only a hidden hearing loss is present [106,107]. The time of day also affects the impact of noise trauma on tinnitus and hearing loss. Nocturnal animals, such as mice, are more sensitive to noise overexposure when the stimulus is applied during the night [108], whereas the converse happens in diurnal animals like gerbils [109], indicating a stronger effect of the auditory insult during the active phase. Interestingly, tinnitus severity in gerbils is highest when the sound overexposure is conducted at the most sensitive time (5 pm), which is also when the largest

noise-induced hearing loss is seen [109]. Ferrets are crepuscular animals, more active at dawn and dusk, and noise overexposure in our experiments was always conducted during the light period. Future studies varying the parameters used for noise overexposure, including the time of day when noise overexposure is applied, will decipher whether it is possible to disentangle hearing loss and tinnitus in the ferret model and its respective interaction with sleep states and circadian rhythms. This study utilised data from only three individual ferrets to explore the relationship between tinnitus and sleep and with no specific control group. However, the study's validity is reinforced by its longitudinal design and the fact that baseline sleep variability in our cases aligns with previously described patterns in ferrets. Moreover, our previous work has shown that similar experimental procedures to those used in this study (intracranial surgery and chronic implants) do not produce behavioural or neuropathological changes [46,47], indicating that the observed changes were most likely caused by the NOE procedure, further supporting our methodology. Future work with larger animal numbers and longitudinal controls will be needed to build on the model and approach we introduce in this study and systematically address the relationship between sleep and tinnitus suggested by our data.

The results of this study point to a potential role for sleep in the transient relief from tinnitus. Increased evoked activity induced by noise overexposure, which is associated with tinnitus [84], was less pronounced during sleep. Therefore, naturally occurring brain states that are known to interfere with sensory signal processing [14,110] may also mitigate the effects of altered excitability following noise trauma. This may be due to sleep of increased intensity after noise overexposure reflecting elevated sleep drive produced by persistent tinnitus–related brain activation in the waking stage [50,51,111–113]. Previous studies indicated that prolonged brain activation raises the internal and network drive of neurons to engage in sleep–specific firing patterns reflected by slow–wave activity [49,114], producing a functional state with the potential to override aberrant brain activity associated with tinnitus [32]. It remains to be seen whether sleep also interferes with aberrant spontaneous activity in individuals with tinnitus.

Tinnitus is the most prevalent sensory phantom percept in humans, but it is not the only one. The phantom limb syndrome in the somatosensory system (the work of Ambroise Paré reviewed in [17], also [16,18]) and the Charles–Bonnet syndrome in the visual system [115] are well recognised phenomena where phantoms are perceived in the absence of the correspondent sensory stimuli. Since sleep attenuates sensory evoked responses (e.g. [116] in the visual cortex, [14] in the perirhinal cortex), sleep–related modulation of sensory phantoms might extend to multiple modalities.

The most widely accepted theory for the basis of phantom percepts, the deprivation theory [117,118], postulates that a reduction in sensory input is an essential trigger, while more recent findings suggest that further changes in sensory precision and predictive coding are necessary for tinnitus to develop [27]. In both models, the most potent risk factors for tinnitus remain clinically identifiable hearing loss, hidden hearing loss [106,119] and subsequent brain plasticity. Therefore, tinnitus provides a novel and potentially useful model to study brain plasticity outside of the homeostatic range and its relationship with hearing impairments. We argue that the dynamics of natural brain states may be a major player in the modulation of either or both conditions and the study of sleep could lead towards new therapeutic avenues in tinnitus, in particular, and in sensory impairments, in general.

## Conclusion

We report the findings of an initial investigation addressing the interaction between sleep and tinnitus in a novel ferret model of noise overexposure–induced tinnitus. A combination of

tinnitus and hearing assessments, vigilance state analysis, and measurement of spontaneous and auditory–evoked EEG activity across vigilance states in three animals suggest a bi–directional interaction between tinnitus and natural brain state dynamics. Individual ferrets developing tinnitus also exhibited sleep impairments, suggesting a link between noise–induced tinnitus and sleep disruption. Neural markers of tinnitus were reduced during sleep, suggesting that the sleep state may transiently mitigate tinnitus. While these results should be considered preliminary, they highlight a new angle to tinnitus research, which could prove fruitful in explaining tinnitus comorbidities and offer opportunities for new experimental approaches.

## Supporting information

**S1 Fig. Assessment of auditory function in ferrets: Baseline assessments and long-term performance in operant gap detection.** (**A**) Baseline performance in the operant gap detection task showing the proportion of left (-30˚) responses by gap length (n = 7). Data encompass all stimuli (broadband noise, BBN, one octave narrow band noise, and one octave narrowband noise, NBN, centred at 1, 4, 8, 16 kHz). Left responses correspond to correct responses for gap stimuli and incorrect responses (false alarm) for no-gap stimuli. Symbols show individual animal means, horizontal lines median values, and boxes interquartile ranges across sessions. Triangle, square and circle symbols represent the animals in which electrophysiological recordings were made during sleep (ferrets 1–3 throughout the manuscript). (**B**) Operant gap detection, proportion of correct responses for each stimulus (1, 4, 8, 16 kHz NBN and BBN). Error bars represent the 95% confidence intervals. Timeline of testing in B and C is colour coded: blue, baseline, red one week after NOE and orange 6 months after NOE. (**C**) Operant gap detection, False alarm (FA) rate by stimulus type across time. FA = 1—proportion of correct responses in no gap trials. Markers represent the mean values from individual animals in panels **A**-**C**. (**D**) Baseline performance in the silence detection task. Proportion of correct responses in BBN, NBN, and silence trials. Grey lines show the mean values for each animal for each stimulus type, and the blue line depicts the mean across animals. Asterisks indicate statistical significance, ** $p < 0.01$, *** $p < 0.001$. (**E**) Exemplar trace of auditory brainstem response (ABR) to 8 kHz centred narrowband noise at baseline. Green markers depict peaks and throughs of ABR waves 1 to 4.
(TIF)

**S2 Fig. Operant gap and silence detection performance and response times.** (**A**) Gap detection performance (left responses) by gap length across time (Baseline, and the two post-NOE assessments, One week and Six months). Left responses are correct responses in gap trials and incorrect responses in no-gap trials. (**B**) Proportion of correct responses by trial type (no gap and gap) across time, colour coded. (**C**, **D**) Response times in the Silence detection paradigm. Response times in NBN, BBN and silence trials for correct responses (**C**) and for incorrect responses (**D**). Asterisks represent statistical significance * $p < 0.05$, ** $p < 0.01$, *** $p < 0.001$. Triangle, square and circle symbols represent the animals in which electrophysiological recordings were made during sleep.
(TIF)

**S3 Fig. Chronic recordings during sleep and wakefulness.** Tinnitus index (TI) and hearing loss relative to baseline (BL) (see Methods). Each panel depicts one ferret. Panels left to right: Ferret 1, Ferret 2, Ferret 3. The tinnitus index (TI) for each animal was based on behavioural impairments in gap and silence detection. Hearing loss was defined as changes in hearing sensitivity (based on ABR thresholds) and changes in ABR magnitude. The numbers and grey colour coding correspond to the size of change relative to BL of the measures at one week and 6

months after noise overexposure.
(TIF)

**S4 Fig. EEG power spectra across vigilance states after noise exposure.** For each ferret, the plots on the upper row show the baseline (BL) and One week EEG spectra, while on the lower row show the BL and Six months EEG spectra for the frontal EEG derivation. Spectra are averages ± standard errors of the mean based on recordings over 48 hours per condition and displayed in 0.5 Hz bins. Originally, average 24 h EEG spectra were produced with a resolution of 0.25 Hz (See Fig 2E). For comparison between conditions, data in each 0.25 Hz bin were merged into 0.5 Hz bins, producing 2 datapoints per 0.5 Hz bin per 24 h recording, and ultimately in 4 datapoints per 0.5Hz bin since EEG spectra obtained during both 24 h periods within the same condition were combined. Two-way ANOVA (factors condition and frequency bin), Tukey's multiple comparisons; horizontal solid black lines over the x axis indicate statistical significance $p < 0.05$.
(TIF)

**S5 Fig. Time course of frontal EEG SWA during NREM sleep.** (BL, blue, One week, red, Six months, orange). NREM slow wave activity (SWA) was calculated per 15-min interval, before being merged into 1-hour intervals, resulting in 4 datapoints per 1-hour bin per 24 hours and 8 datapoints per condition (consisting of two 24h recording periods). Shaded error bars are standard deviations. Two-way ANOVA (factors condition and time interval), Tukey's multiple comparisons; black square markers indicate differences at $p < 0.05$. Green squares in panels A and B mark time intervals with less than 3 data points in either condition. Times are expressed in zeitgeber time (ZT), time 0 corresponds to the start of the light period.
(TIF)

**S6 Fig. Definitions of early, mid and late components of the AER and AER magnitudes.** Each row depicts one ferret (ferrets 1–3). Left column (**A,C,E**) time windows for response components. Average occipital EEG evoked potential during BL recordings with an outline of the time windows where the response components (R1, R2, R3) were defined. Similar windows were applied for the frontal EEG signal (not shown). Note that a different number of response windows was defined for different animals. Right column (**B,D,F**) Definition of response magnitudes. A custom MATLAB algorithm selected the maximum value within each response window and the subsequent minimum to compute the response magnitude for each response component.
(TIF)

**S7 Fig. EEG auditory evoked responses across vigilance states (Ferret 1).** Auditory evoked responses for EEG frontal (top panels) and EEG occipital derivations (bottom panels) during the BL (baseline), 1 week and 6 months conditions. The dotted red vertical line in each panel depicts the stimulus onset. The averages displayed in this figure are based on snippets of raw recording data after the exclusion of signal artefacts (see Methods: vigilance state scoring) and are averages over signals for all stimulus types (820 ms narrow-band sounds centred around 1, 4, 8 and 16 kHz with a 38 ms silent gap at the centre of the stimulus,) and multiple levels (40, 50, 60 and 65 dB SPL).
(TIF)

**S8 Fig. EEG auditory evoked response across vigilance states (Ferret 2).** Auditory evoked responses for EEG frontal (top panels) and EEG occipital derivations (bottom panels) during the BL (base line) and 1 week condition. The signal quality in 6 months condition was insufficient for quantitative analysis. The dashed red vertical line in each panel depicts the stimulus

onset. The averages displayed in this figure are based on snippets of raw recording data after the exclusion of signal artefacts (see Methods: vigilance state scoring) and are averages over signals for all stimulus types (820 ms narrow-band sounds centred around 1, 4, 8 and 16 kHz with a 38 ms silent gap at the centre of the stimulus,) and multiple levels (40, 50, 60 and 65 dB SPL).

(TIF)

**S9 Fig. EEG auditory evoked responses across vigilance states (Ferret 3).** Auditory evoked responses for EEG frontal (top panels) and EEG occipital derivations (bottom panels) during the BL (baseline) and one week condition. The signal quality in six months condition was insufficient for quantitative analysis. The dashed red vertical line in each panel depicts the stimulus onset. The averages displayed in this figure are based on snippets of raw recording data after the exclusion of signal artefacts (see Methods: vigilance state scoring) and are averages over signals for all stimulus types (820 ms narrow-band sounds centred around 1, 4, 8 and 16 kHz with a 38 ms silent gap at the centre of the stimulus,) and multiple levels (40, 50, 60 and 65 dB SPL).

(TIF)

**S10 Fig. AER magnitude for each stimulus.** Averages across all vigilance states for ferrets 1–3 organised in columns and for the different EEG configurations, (**A**-**C**) Frontal EEG evoked response (AER) for all stimuli before (baseline, blue) and after NOE (1 week, red; 6 months, orange) and (**D**-**F**) for occipital EEG. Data are bootstrapped averages across response components, vigilance state and sound level (see Methods for details). Error bars are standard errors. All group comparisons differed at $p<0.001$ (***, GLMM, factor 'condition'). AER magnitude for each stimulus. Averages across all vigilance states. Each panel depicts one ferret. Panels left to right: Ferret 1, Ferret 2, Ferret 3. All group comparisons differed at $p<0.001$ (GLMM, factor 'condition').

(TIF)

**S11 Fig. Grand average of evoked responses for Ferret 1 (baseline -BL, 1 week, 6 months) and by recording site.** Number of trials contributing to the average are indicated above each panel. Note that trials falling into epochs scored as 'artefact' during the manual scoring procedure were not included.

(TIF)

**S12 Fig. Grand average of evoked responses for Ferret 2 (baseline -BL, 1 week, 6 months) and by recording site.** Number of trials contributing to the average are indicated above each panel. Note that trials falling into epochs scored as 'artefact' during the manual scoring procedure were not included. To produce this figure and because signals in the six months condition were generally of lower quality, only trials where the signal's standard deviation did not exceed double the average standard deviation were included.

(TIF)

**S13 Fig. Average evoked responses by condition (baseline -BL, 1 week, 6 months) for Ferret 3 and by recording site.** Number of trials contributing to the average are indicated above each panel. Note that trials falling into epochs scored as 'artefact' during the manual scoring procedure were not included. To produce this figure and because signals in the Six months condition were generally of lower quality, only trials where the signal's standard deviation did not exceed double the average standard deviation were included.

(TIF)

**S14 Fig. Changes in evoked responses after noise exposure during sleep and wake.** Values are average auditory evoked response magnitudes as a percentage of the baseline condition for the frontal EEG signal (left panels) and for the occipital EEG signal (right panels). In each panel, data are averaged across stimulus levels and response windows. The x-axis represents stimuli, the y-axis represents the vigilance states (Wake, NREM, REM, REM2). The first two rows in the figure (consisting of 4 panels) correspond to Ferret 1, the third row to Ferret 2, and the last row to Ferret 3. For all animals average auditory evoked response magnitudes for 1 week were calculated. In addition, magnitudes for 6 months were only calculated for Ferret 1 and are shown in the second row of figure panels.
(TIF)

**S15 Fig. Increased cortical spontaneous activity after noise overexposure.** (**A**) Spontaneous activity of acoustically-responsive neurons for a Control ferret (n = 543 units) and two NOE animals Ferret 2 (n = 237) and Ferret 3 (n = 294). NOE animals exhibited significantly higher spontaneous activity (Kruskal-Wallis, $\chi^2_{(2,1073)}$ = 122.83; p<0.0001) and a greater variability as indicated by the larger interquartile range. In each box, the central mark indicates the median, and the bottom and top edges of the box indicate the 25th and 75th percentiles, respectively. The whiskers extend to the most extreme data points not considered to be outliers, and the outliers are plotted individually using the '+' marker symbol. (**B**) Cumulative distribution of cortical units according to their spontaneous firing rates for the three ferrets. Cumulative functions for NOE ferrets showed a broader distribution skewed towards higher rates. (**C**) Difference between evoked (to broadband noise presentation) and spontaneous interspikes intervals (ISI). Although a reduction of the ISI with stimulus presentation was expected because of increased activity following the stimulus, this reduction was significantly more marked in the control ferret than after NOE (Kruskal-Wallis, $\chi^2_{(2,629)}$ = 28.97; p<0.0001), suggesting that the temporal properties of the responses was less precise in NOE ferrets[a].
(TIF)

# Acknowledgments

We are grateful to Susan Spires and Ana Sánchez Jimenez for their support in conducting ferret operant behaviour experiments and to Sara Rolle Sonora and Keilina Monteiro de Canto who contributed to the initial spike sorting from the terminal electrophysiological recordings.

# Author Contributions

**Conceptualization:** Linus Milinski, Fernando R. Nodal, Andrew J. King, Vladyslav V. Vyazovskiy, Victoria M. Bajo.

**Data curation:** Linus Milinski, Fernando R. Nodal, Matthew K. J. Emmerson, Victoria M. Bajo.

**Formal analysis:** Linus Milinski, Fernando R. Nodal, Matthew K. J. Emmerson.

**Funding acquisition:** Andrew J. King, Victoria M. Bajo.

**Investigation:** Linus Milinski.

**Methodology:** Linus Milinski.

**Project administration:** Victoria M. Bajo.

**Resources:** Andrew J. King, Victoria M. Bajo.

**Supervision:** Fernando R. Nodal, Vladyslav V. Vyazovskiy, Victoria M. Bajo.

**Writing – original draft:** Linus Milinski, Victoria M. Bajo.

**Writing – review & editing:** Linus Milinski, Fernando R. Nodal, Andrew J. King, Vladyslav V. Vyazovskiy, Victoria M. Bajo.

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
