## [Decision Letter · Decision Letter 0]

4 Jun 2024

PONE-D-24-17819

Cortical evoked activity is modulated by the sleep state in a ferret model of tinnitus. A case study.

PLOS ONE

Dear Dr. Bajo,

Thank you for submitting your manuscript to PLOS ONE. After careful consideration, we feel that it has merit but does not fully meet PLOS ONE’s publication criteria as it currently stands. Therefore, we invite you to submit a revised version of the manuscript that addresses the points raised during the review process.

Your manuscript has been reviewed by two experts in the topic. As you will see below, while the reviewers found the idea of examining the relationship between tinnitus and sleep to be timely and interesting, they both felt that the claims made in the paper are only weakly supported by the data, and I agree. Specifically, there appear to be two main issues: First, as pointed out in Review #1, the lack of a sham control group, in combination with the high variability in the very small group of only 3 animals with chronic recordings, makes it difficult to assess whether a link between tinnitus and sleep disruption exists in the data. Second, as pointed out in Review #2, there are additional concerns with the validity of the behavioral tinnitus indices as no physiological correlates of tinnitus are reported, interindividual differences in responsiveness to NOE are not sufficiently considered, and ABR thresholds indicate an unusual progressive hearing loss that makes it difficult to distinguish effects of threshold shifts and tinnitus. It might prove difficult to fully address these concerns in a revision of the paper, but I would leave this decision up to you. Should you submit a revision, I will likely send out the new version to the same reviewers. An alternative route forward, as pointed out by Reviewer #2, could be to split the establishment of the new ferret model of tinnitus and the examination of sleep effects into separate manuscripts, which might then be more appropriate as a new submission rather than as a revision.

We look forward to receiving your revised manuscript.

Kind regards,

Patrick Bruns

Academic Editor

PLOS ONE

Journal Requirements:

2. In order to comply with PLOS ONE's guidelines for animal experiments (https://journals.plos.org/plosone/s/submission-guidelines#loc-animal-research), please specify the disposition of animals at the end of the study (e.g. euthanasia, returned to home colony, etc.). If animals were euthanized following the study, please provide the method of sacrifice and and describe any efforts that were undertaken to reduce animal suffering.

   "Royal National Institute for Deaf People (RNID, Grant S52_Bajo). VMB

    Wellcome Trust [WT108369/Z/2015/Z]. AJK"

Reviewers' comments:

Reviewer's Responses to Questions

**Comments to the Author**

1. Is the manuscript technically sound, and do the data support the conclusions?

Reviewer #1: Partly

Reviewer #2: Partly

2. Has the statistical analysis been performed appropriately and rigorously? 

Reviewer #1: Yes

Reviewer #2: No

3. Have the authors made all data underlying the findings in their manuscript fully available?

Reviewer #1: Yes

Reviewer #2: No

4. Is the manuscript presented in an intelligible fashion and written in standard English?

Reviewer #1: Yes

Reviewer #2: Yes

5. Review Comments to the Author

Reviewer #1: In this study, Milinski et al establish a behavioral model of noise-overexposure induced tinnitus in the adult female ferret, as verified by operant gap detection and silence detection paradigms. They combine their behavioral testing with chronic electrophysiological recordings to measure vigilance states before and after noise overexposure and relate the results to the behavioral evidence of tinnitus and hearing impairment. The premise of the study, which is that (1) the ferret, given its lower frequency hearing range compared to mice, long lifespan and ability to perform complex behavior tasks, is a valuable model for human subjective tinnitus, and (2) the ferret is also amenable to chronic electrophysiological and behavioral testing of sleep/wake states, is an appropriate model to investigate the oft-reported connection between tinnitus and sleep disturbance in humans, is sound. The topic is of broad interest to clinicians and basic researchers studying both sleep and sensory disorders, and the mechanistic connections between hearing impairment and sleep are understudied despite emerging attention focused on sound-induced recruitment of arousal systems in wakefulness. Therefore, the study is timely and interesting.

The authors highlight the bi-directional interaction between tinnitus and sleep, which while intriguing, is not strongly supported by the evidence. A major reason for this is the variability in the admittedly small group of animals tested. There is certainly value in testing a smaller cohort of animals over long time periods, but part of this strength is leveraged by testing multiple times at baseline prior to NOE. A major weakness of the study is the inability of the reader to understand the variability expected in sleep architecture, tinnitus behavior, and EAPs at baseline pre-exposure, in addition to the variability that could be expected over a 6 month testing period in a sham-exposed animal. A sham group would experience similar levels of signal degradation from an electrode, behavioral variability, and stress due to surgical or exposure procedures, but presumably without tinnitus.

Therefore, in order to claim a link between tinnitus and sleep disruptions, the authors should provide an analysis of baseline variability due to real underlying variability and/or experimental noise. Another way to address this concern would be to point to other studies of ferret sleep architecture that may give a basis for comparison, if there are any. If these elements are not included, the authors should instead adjust their claims accordingly.

Other comments and questions listed below:

1. Was the position of the ferret taken into consideration when presenting stimuli during sleep? In other words, how was an even sound field constructed so that the SPL would not vary at different locations in the chamber? I imagine at lower frequencies this would not be much of an issue, but perhaps at 16kHz, for example, there could be large differences in SPL at different locations in the chamber.

2. Related to the first point, why was unilateral NOE used, and if animals slept with one ear down by the floor, was there a requirement that the exposed versus intact ear would be facing the speaker?

3. The tinnitus index calculation is a bit confusing-- 1/3 of the index is based on performance at detecting a continuous sound, which is added to the other metrics and not weighed differently. This seems to be a readout of behavioral performance and hearing sensitivity, rather than tinnitus per se. Why is it added to the other values rather than used to normalize or otherwise scale the values?

4. The baseline tinnitus index (pre-exposure) should be shown in figure 3A.

5. The EEG recordings, at least in case 2 & 3, appear to degrade significantly in terms of quality by the 6 month time point. How do the authors verify the accuracy of their sleep stage scoring results given the increase in noise?

6. The discussion could be enhanced by discussing the findings in humans with tinnitus, as a few polysomnography studies have indicated increased sleep latency and decreased deep NREM and REM as potentially linked with the condition. This could serve as a point of comparison to the ferret results.

Reviewer #2: General

This study attempts first to develop a new model of tinnitus in ferrets, and second to study the effects of sleep. These would be better separated into two papers, enabling the authors to more thoroughly develop the model and establish its validity before going on to study the effects of sleep,

In the first part, while ambitions and meritorious in its scope of having three separate behavioral measurements of tinnitus, there are some factors that decrease enthusiasm for the application of these methods. First, only a percentage of animals (and people) develop tinnitus after noise over exposure. Thus, it would be important to show individual data for each animal rather than just mean results. Second, correlating the behavior with a physiological correlate of tinnitus (eg increased spontaneous rates or synchrony in an auditory structure) would add to the validity of these behavioral measures.

Enthusiasm for the results of the study is diminished considering the progressive increase in ABR thresholds observed over the 6 months. This is problematic as it is now difficult to really distinguish the effects of hearing loss (threshold shifts) and tinnitus. It is surprising that there is a progressive threshold increase in ferrets at 6 months as other studies in animals have shown temporary threshold shifts with this level of noise exposure delivered under anesthesia.

The idea of examining the relationship of tinnitus to sleep is interesting. Strengthening the theoretical link at the outset would improve the impact of the paper. The same limitations would apply to this second part of the paper as hearing loss would likewise affect the interpretation of tinnitus.

Specific

Abstract:

Rationale for exploring sleep and tinnitus not made clear in abstract – should make this clearer.

Rather state no available cure as there are effective treatments in recent developments.

Introduction

First paragraph is very complex and difficult to follow. Suggest simplifying.

Second paragraph provides a different rationale and aim for the study – ie to develop a new animal model for tinnitus. Perhaps soften the wording here – eg use a different model to study sleep-tinnitus interactions.

Methods

Only 3 animals were used for chronic recordings. It would be good to have more animals with and without tinnitus without the confound of hearing loss. Consider developing the model to produce a temporary theshold shift instead of permanent.

Tinnitus measurements – three behavioral measurements is a strength. However, previous gap detection paradigms were of highest value when showing correlations with physiological states that indicate tinnitus (eg increased spontaneous rates and synchrony of an auditory region). Is there some activity that could serve this purpose in the present study?

Please provide more rationale for assessing ‘vigilance state’ from frontal and occipital regions.

It is stated that silent gap detection decreased after noise exposure. The results show mean increases in gap detection. This is problematic that individual animals are not shown as previous studies have shown that only about half of animals develop tinnitus, therefore it is not valid to show mean data. It would be better to separate the animals into those that develop tinnitus and compare them with those that do not develop tinnitus. The same is true fo the ABR results. Individual animals should be shown in addition to mean data.

In addition, it is problematic that the ABR thresholds increase progressively at one week and 6 months (unless the figure is incorrect). It would be helpful if the colors for 6 months and one week were more distinctive. But assuming the figure is correct, this means that the animals had hearing deficits of more than 40 dB at some frequencies. This also means that some of the gap detection results could be due to hearing loss and not to tinnitus. It is also unusual for animals to show worsening of ABR thresholds at 6 months for this level of noise exposure. Usually this produces a temporary threshold shift and not a permanent threshold shift as in this case.

Discussion

As the study is very long, it is suggested to divide this paper into two papers with the tinnitus model being the first paper. The discussion is very long and rambling and would be better focused specifically on the results of this study rather than numerous speculations regarding etiology, hyperacusis and other models.

6. PLOS authors have the option to publish the peer review history of their article (what does this mean?). If published, this will include your full peer review and any attached files.

Reviewer #1: No

Reviewer #2: No

---

## [Author Response · Author response to Decision Letter 0]

27 Jul 2024

Oxford, July 27, 2024

Dear Dr Bruns, PLoS One Editor,

Re: PONE-D-24-17819 “Cortical evoked activity is modulated by the sleep state in a ferret model of tinnitus. A case study.”

We are pleased that you are interested in publishing our study in the Journal and we appreciate the reviewers’ feedback. We have worked hard to address every point raised by the reviewers and believe we have produced a more robust and much improved paper that will now be ready to be accepted. 

We have responded to each point raised by you and by the reviewers. Any revisions made to the manuscript text have been highlighted in red fonts when quoted in this cover letter, the response to the reviewers, and also in the revised manuscript itself. Editor and Reviewers’ comments have been kept in blue, our responses are in black. The line numbers mentioned here correspond to the line numbers in the initial submission.

With this letter, we submit a marked-up copy of our manuscript that highlights in red changes made to the original version 'Revised Manuscript with Track Changes' and unmarked version of our revised paper without tracked changes labelled 'Manuscript'.

Dear Dr. Bajo,

Thank you for submitting your manuscript to PLOS ONE. After careful consideration, we feel that it has merit but does not fully meet PLOS ONE’s publication criteria as it currently stands. Therefore, we invite you to submit a revised version of the manuscript that addresses the points raised during the review process.

Your manuscript has been reviewed by two experts in the topic. As you will see below, while the reviewers found the idea of examining the relationship between tinnitus and sleep to be timely and interesting, they both felt that the claims made in the paper are only weakly supported by the data, and I agree. Specifically, there appear to be two main issues: First, as pointed out in Review #1 the lack of a sham control group, in combination with the high variability in the very small group of only 3 animals with chronic recordings, makes it difficult to assess whether a link between tinnitus and sleep disruption exists in the data. (Reviewer 1) 

#E1 

We thank the Editor and Reviewer 1 for highlighting these important considerations.

Generally, in ferrets, longitudinal studies are often used to establish within comparisons and reduce the number of animals, in line with the 3Rs principle. This information and the advantages of this model has been reflected in the manuscript, at the end of the introduction, in the methods, and at the beginning of the discussion.

In particular, we do not use a sham control group in this study, which is one of its main limitations and an obvious step for future projects. However, the aim of this study was to set up a new animal model for chronic tinnitus and for researching its relationship with sleep. An important part of this was to characterise tinnitus in a detailed manner an in a longitudinal design, which we therefore prioritised over a high animal number. As stated in our response to reviewer 2, we believe that this work, while being a case study with inherent limitations on the strength of general interpretations, it is an important contribution to the tinnitus field as it for the first time proposes, tests and uses an empirical approach to systematically study the connection between chronic tinnitus and sleep. It will set the foundation for a new research angle important for furthering our understanding of tinnitus and the development a possible new treatment angle. 

To account for individual variability in sleep-wake distribution, we have analysed in the revised version the baseline variability in the ferrets’ sleep pattern as suggested by reviewer #1 (R1#3) and we show that it closely matches that reported in previous studies where ferret sleep was investigated and sleep was scored by similar means (based on EEG/EMG signals). For more details, see R1#3.

We added the following section to the manuscript:

Line 655: 

The animals spent most of the time (75.9±4.86%) sleeping in undisturbed baseline conditions (71.5%, 85.6%, and 70.6%, Fig. 6A–C), similar to previously reported sleep durations in ferrets (70.34±1.69%, [41]). Durations in vigilance state durations were similar across days in each ferret (Fig. 2F) and variability between animals also closely matched previous reports [41]).

We would like to add that although ferrets present individual variability in their behaviour, the behaviour does not usually degrade over time. Our Home Office animal licence allow us to run behavioural testing in adult ferrets for 30 months, which starts when they are young adults, about 1 year old. We have shown that behavioural performance changes little over this long period and is not affected by long, even consecutive surgeries. For example, in Bajo et al. (2010) we used two consecutive cranial surgeries under general anaesthesia to selectively ablate auditory corticocollicular neurons, and showed with specific controls having the same surgery but in different brain areas, that the surgery did not produce any deficits in sound localization accuracy (Bajo et al., Nat Neurosci. 2010 Feb;13(2):253-60. doi: 10.1038/nn.2466. See Fig. 1 and methods). Another example is Bajo et al. (Nat Commun. 2019 Jul 12;10(1):3075. doi: 10.1038/s41467-019-10770-4), where we tested sham-operated animals for 2 years in sound localization paradigms with and without earplugs and saw no degradation in their ability to adapt to the altered spatial cues resulting from unilateral earplugging over long periods of testing (see Bajo et al. 2019, Supplementary Figure 4, n=13). In addition, experimental cases with intracranial surgery under general anaesthesia to implant fibre optic devices for 30 months (n=13) and multiple periods of behavioural testing showed no changes in their health and consistently normal sound localization other than a specific impairment in earplug adaptation when the auditory cortex was optogenetic silenced (see Bajo et al., 2019, Supplementary Figure 5).

In the revised version of the manuscript, we added a paragraph to the methods to specify that while no specific controls have been used in this study, the controls and experimental groups used in several previous studies showed no indication of any alteration in behaviour due to cranial surgery under general anaesthesia or prolonged behavioural testing. 

Line 109. “The longitudinal design of this study spanning more than 9 months allowed each animal to act as its own control by comparing the results before and after noise overexposure. We did not use specific controls comprising implanted animals without noise overexposure, since previous ferret experiments show no changes in behavioural testing or neural recordings following similar surgical procedures and/or intracranial implantation (e.g. Bajo et al., 2010; 2019). In addition, one separate, naïve animal was used as a control to compare physiological correlates of tinnitus with two out of the three ferrets implanted for chronic recordings (see methods).” 

Second, as pointed out in Review #2, there are additional concerns with the validity of the behavioral tinnitus indices as no physiological correlates of tinnitus are reported, interindividual differences in responsiveness to NOE are not sufficiently considered, and ABR thresholds indicate an unusual progressive hearing loss that makes it difficult to distinguish effects of threshold shifts and tinnitus. 

#E2 To date no a definitely behavioural or physiological correlate to tinnitus has been established. In humans the only accepted prove of tinnitus is its subjective reporting, the best proxy we have for it in animal models is the observed changes in auditory behaviours that are susceptible of being altered by the phantom perception. We think therefore that the changes we reported and summarize in the tinnitus indexes are a valid estimation of tinnitus suffering.

However, to address the concern regarding the validity of the behavioural tinnitus indices, we have further strengthened our study by incorporating an additional physiological correlate of tinnitus. Specifically, we conducted recordings using Neuropixels probes in the auditory cortex of two out of three implanted ferrets, as well as in one sham-operated control for comparison. Initially, we had planned to present these data in a future publication; however, in response to your suggestion, we have now included them in the revised manuscript. These recordings were performed under general anaesthesia following the completion of the experimental protocol. Briefly, we found that both assessed NOE animals exhibited a significantly greater median spontaneous activity than units recorded in the control case (Kruskal-Wallis, χ2 (2,1073) = 122.83; p<0.0001) (new supplementary Fig 16A), as well as shortened interspike intervals (ISIs). Although we cannot fully untangle the contribution of this change in excitability to the reduction of the ISI, it is worth noting that such shortening was more pronounced in control units than after NOE. When considering together the observed differences in spontaneous activity between cases and the less marked changes in the ISI, both are suggestive of a reduced temporal precision in the cortical activity of the NOE animals, which is consistent with the behavioural changes exhibited by those animals. 

The methods and results for this experiment have been included in the main manuscript, and a new supplementary figure added (see below).

Line 480:

Final recordings with Neuropixels probes in the Auditory Cortex

In two out of the three implanted cases (Case 2 and Case 3) and in one additional control, we performed terminal recordings under general anaesthesia in the left auditory cortex (contralateral to the noise overexposured ear) using Neuropixels probes. 

General anaesthesia was induced and maintained using Domitor (0.022 mg/kg/h) and Narketan 10 (5 mg/kg/h) in 0.9% sodium chloride, pH 7.2-7.4 with 5% glucose delivered intravenously (3-5 ml/h) for the complete duration of the recordings (24-36 hours). After tracheal intubation, the animal was artificially ventilated. Atropine sulphate (0.06 mg/kg s.c.), doxapram hydrochloride (4 mg/kg, s.c.), dexamethasone (0.5 mg/kg s.c.) and viscotears were administer every 6 hours to minimize pulmonary secretion, reduce respiratory depression, prevent cerebral oedema, and protect the eyes, respectively. Perioperative analgesia and antibiotics were also administered during the surgery as mentioned for the previous surgery, and depth of anaesthesia, respiratory rate, ECG, and end-tidal CO2 were monitored. Rectal temperature was maintaining at 38 °C with the help of a homeothermic electric blanket and a forced air-warming system. 

After removing the chronic implant in the NOE animals, a craniotomy (4x4 mm2) was performed over the ectosylvian gyrus to reveal the auditory cortex. The animal was placed in an anechoic chamber and the dura mater was opened to visualize the primary auditory cortex.

Neural recordings were performed under anaesthesia (mixture of Domitor (medetomidine hydrochloride, 0.022 mg/kg, Orion Pharma; and Ketaset (ketamine hydrochloride, 5 mg/kg, Fort Dodge Animal Health) using Neuropixels 3A probes (single shank with 384 channels) and streamed to disk with SpikeGLX (https://billkarsh.github.io/SpikeGLX/) (sampling rate 25 kHz and filtered between 300 and 3000Hz). Spike clusters were identified using Kilosort2 spike sorting algorithm (Pachitariu et al 2016), curated by Phy (https://github.com/kwikteam/phy) and further analyzed using custom-written scripts in Matlab (R2023a). We analyzed the spontaneous firing behaviour of clusters that were responsive to broadband noise (duration 300ms, intensity 60-90 dB SPL), as defined by spike counts in the first 100 ms window (1ms bins) following the stimulus presentation being significantly different from a window of the same duration preceding the stimulus (paired t-tests, P<0.05).

Line 862: 

Neuropixels Recordings in the Auditory Cortex

In two out of the three implanted cases (Case 2 and Case 3) and in one additional control, we performed final recordings in the left auditory cortex (contralateral to the noise overexposured ear). Acoustically-responsive cortical units recorded in NOE Cases 2 (median/interquartile 0.958/5.527) and 3 (md/iqt 2.713/5.764) exhibited a significantly greater median spontaneous activity than units recorded in the control case (md/iqt 0.275/1.975) (Kruskal-Wallis, χ2 (2,1073) = 122.83; p<0.0001) (supplementary Fig 16A). The cortical units also had a broader distribution of firing rates after NOE than in the control animal (supplementary Fig 16B), indicating that the observed increase in median spontaneous firing rate was due to an overall increase with fewer units exhibiting a low spontaneous firing rate. Bursting activity was very low for all cases (control: 0.025 Hz, case 2: 0.012Hz and case 3: 0.004 Hz) with only 12.6% (30/237 units) in case 2, 5.4% (16/294 units) in case 3 and 9.6% (52/543 units) in the control case producing at least one burst of spikes. 

To gain an insight into the temporal precision of the cortical responses, we calculated for each unit the change in the interspike intervals (ISI) during their evoked activity (100ms after stimulus presentation) compared to their spontaneous activity. Because we only analyzed acoustically-responsive units, the increased number of spikes would result in shorter ISI and therefore negative differences as was observed for all three cases, control and NOE (supplementary Fig 16C). Although we cannot fully untangle the contribution of the change in the excitability of the units to the reduction in the ISI, it is worth noting that such shortening was more pronounced in control units than after NOE. The observed differences in spontaneous activity between cases and less marked changes in the ISI are together suggestive of reduced temporal precision in the cortical activity in the NOE animals, which is consistent with the behavioural changes exhibited by those animals. 

Line 1299:

Fig S16. Increased cortical spontaneous activity after noise overexposure. (A) Spontaneous activity of acoustically-responsive neurons for a control case (n= 543 units) and two NOE animals Case 2 (n=237) and Case 3 (n=294). NOE animals exhibited significantly higher spontaneous activity (Kruskal-Wallis, χ2 (2,1073) = 122.83; p<0.0001) and a greater variability as indicated by the larger interquartile range. In each box, the central mark indicates the median, and the bottom and top edges of the box indicate the 25th and 75th percentiles, respectively. The whiskers extend to the most extreme data points not considered to be outliers, and the outliers are plotted individually using the '+' marker symbol. (B) Cumulative distribution of cortical units according to their spontaneous firing rates for the three cases. Cumulative functions for NOE cases showed a broader distribution skewed towards higher rates. (C) Difference between evoked (to broadband noise presentation) and spontaneous interspike intervals (ISI). Although a reduction of the ISI with stimulus presentation was expected because of increased activity following the stimulus, this reduction was significantly more marked in the control case than after NOE (Kruskal-Wallis, χ2 (2,629) = 28.97; p<0.0001), suggesting that the temporal properties of the responses was less precise in NOE cases.

It might prove difficult to fully address these concerns in a revision of the paper, but I would leave this decision up to you. Should you submit a revision, I will likely send out the new version to the same reviewers. An alternative route forward, as pointed out by Reviewer #2, could be to split the establishment of the new ferret model of tinnitus and the examination of sleep effects into separate manuscripts, which might then be more appropriate as a new submission rather than as a revision.

#E3 (also R2#1 & R2#13). We thank the Editor and Reviewer for the suggestion to separate this study into two papers, the first one about the ferret as animal model of tinnitus, and the second one studying the interaction between tinnitus and sleep. We know that we have strong data for the first part o

---

## [Decision Letter · Decision Letter 1]

17 Sep 2024

PONE-D-24-17819R1Cortical evoked activity is modulated by the sleep state in a ferret model of tinnitus. A case study.

PLOS ONE

Dear Dr. Bajo,

Thank you for submitting your manuscript to PLOS ONE. After careful consideration, we feel that it has merit but does not fully meet PLOS ONE’s publication criteria as it currently stands. Therefore, we invite you to submit a revised version of the manuscript that addresses the points raised during the review process.

**First of all, let me apologize for the delay in getting back to you about your submission. Unfortunately, one of the original reviewers was no longer available. Therefore, the revised manuscript has been reviewed by original Reviewer #1 as well as two new reviewers. As you will see below, while the reviewers found the manuscript to be improved, they all had remaining major concerns. Parts of these concerns are due to oversights on your part, such as missing main figures and an incomplete reference list. I agree with the reviewers that additional work is required to improve the figures (Review #1) and the readability of the introduction and discussion (Review #3). Moreover, conclusions are not fully supported by the data and should be toned down accordíngly (Review #4). However, the reviewer comments leave me with enough hope that the remaining concerns can be addressed in another major revision of the manuscript.**

We look forward to receiving your revised manuscript.

Kind regards,

Patrick Bruns

Academic Editor

PLOS ONE

Reviewers' comments:

Reviewer's Responses to Questions

**Comments to the Author**

1. If the authors have adequately addressed your comments raised in a previous round of review and you feel that this manuscript is now acceptable for publication, you may indicate that here to bypass the “Comments to the Author” section, enter your conflict of interest statement in the “Confidential to Editor” section, and submit your "Accept" recommendation.

Reviewer #1: (No Response)

Reviewer #3: (No Response)

Reviewer #4: (No Response)

2. Is the manuscript technically sound, and do the data support the conclusions?

Reviewer #1: Yes

Reviewer #3: Yes

Reviewer #4: Partly

3. Has the statistical analysis been performed appropriately and rigorously? 

Reviewer #1: N/A

Reviewer #3: Yes

Reviewer #4: Yes

4. Have the authors made all data underlying the findings in their manuscript fully available?

Reviewer #1: Yes

Reviewer #3: Yes

Reviewer #4: Yes

5. Is the manuscript presented in an intelligible fashion and written in standard English?

Reviewer #1: Yes

Reviewer #3: Yes

Reviewer #4: Yes

6. Review Comments to the Author

**Reviewer #1: **I thank the authors for addressing my comments and adding valuable data to the manuscript as well as editing the text for clarity. I have a few remaining suggestions, the overall comment remaining being that the manuscript would benefit greatly from another round of editing on the figures.

-The manuscript can be hard to digest given that it explores the broad topics of tinnitus and sleep, and as a case study requires the reader to consider each individual rather than a big-picture result. As the authors chose to keep the paper together rather than splitting it as R2 suggested, to improve readability I would suggest moving a couple items from the supplemental material to the main figures:

-One of the main results highlighted in the discussion, that is consistent across the 3 animals -- an increase in REM2-- is not explicitly shown in a figure (the REM2 amounts are shown in separate figures for the baseline and post-NOE conditions). Pre- and post- NOE REM2 measurements (along with the amounts of other sleep stages) should instead be side-by-side in a figure, with error bars, and statistical analysis if possible.

-The behavioral schematics from figure s1 should be moved to figure 1 as this adds valuable context.

-In general, the figures should be edited for visibility. In places the font sizes are different across panels, e.g. figures 3 and 4. In Figure 3B, middle, there are some graphics that appear to be a screenshot artifact that should be removed.

-The components that make up the tinnitus index calculation, and the assumptions of the TI, should be described verbally in the results section, as this gives important context to the data shown in figures.

-In the discussion, lines 897-898 contain a typo.

**Reviewer #3: **The authors present behavioral and electrophysiological data investigating the link between tinnitus and sleep in 7 adult female ferrets, with 3 animals receiving long term EEG recordings and from 2 (+1) animals, auditory cortex activity was recorded with sharp electrodes. This resulted in three single cases, that could be described over six months in great detail. The manuscript is already revised and most concerns regarding the methodology and results have been accounted for.

Overall I suggest a major revision, especially in the Introduction and Discussion sections, in the latter many references (after 68) are missing.

Major points:

One major point that I am missing from the manuscript is the introduction to the different neurophysiological models of tinnitus development and the introduction to sleep and sleep related measurements (polysomnography). Both are relevant for the understanding of the used methods and the interpretation of the results.

For the reader, it would improve the readability, if instead of post1 and post2 the real timepoints are presented in the text and Figures. In some of the supplements, this is already the case and improves the intelligibility significantly.

L888ff: Add here also the development aspect from the introduction. This is relevant, as you have early and late recording timepoints.

Also compare your behavioral model of tinnitus identification to other behavioral animal models, including conditioning and reflexive models.

Please check for the missing references after 68. Please also increase your comparisons of ABR shifts to more than one reference (69) for animals and humans (70).

For all references after 68, I cannot decide if they are appropriate or missing points. Please double-check.

L 955: An important tinnitus animal model is the Mongolian gerbil, please add references here.

L964ff: Here you could speculate about the circadian influence on hearing impairment. In rodents, it has been shown to be most relevant for hearing loss and tinnitus development: DOI 10.3389/fnins.2022.830703; 10.1121/1.5132290

L967: reference style changed here. For hidden hearing loss, also DOI 10.1111/ejn.15334 is important.

Minor points:

L XXX: remove “we”

L XXX: “no gap” should be “no-gap”

L 656: Fig. 6A-C should be Fig. 2A-C

**Reviewer #4: **This is an interesting paper and would be of interest to many tinnitus researchers. However the data are over-interpreted. The authors have made changes to respond to reviewers but need to make additional changes in order for the paper to be acceptable for publication.

My biggest problem with the data from the 3 animals is that there is no consistent pattern (see also fig S4) at any of the sleep states and the study lacks sufficient number of animals to say anything meaningful. It does show that these studies are possible and it warrants further investigation but authors should temper conclusions throughout the manuscript.

Abstract and throughout: As the 3 animals that were investigated for sleep patterns all had tinnitus the statement that “Animals that developed signs of tinnitus consistently developed sleep impairments, suggesting a link between the emergence of noise–induced tinnitus and sleep disruption” is an overstatement. It may be the hearing loss that causes the problem not tinnitus.

It should also be made clear in the abstract that only 3 animals were investigated.

Introduction:

It is mentioned that “We tracked behavioural markers of noise–induced tinnitus and hearing impairment in a ferret model over a period of six months after noise overexposure” This suggests regular testing whereas testing was only done at two timepoints. This should be made more clear. In addition only 3 of the animals were tested at the 6 months timepoint. This is mentioned in results but not made clear in introduction or methods.

Same as Abstract comments: “Animals developing tinnitus also exhibited sleep impairments, suggesting a link between the emergence of noise–induced tinnitus and sleep disruption.” This is misleading as no animals without tinnitus were investigated for sleep patterns.

Methods:

New figure 6 was not available and also no figure 5? Although there is no reference to a figure 5?

The additional data and figure showing spontaneous firing rate increase in the tinnitus animals versus naïve control is useful but I suggest the authors reference “Meredith MA, Keniston LP, Allman BL. Multisensory dysfunction accompanies crossmodal plasticity following adult hearing impairment. Neuroscience. 2012 Jul 12;214:136-48” who showed this as well following hearing loss and therefore it cannot be stated this is a marker for tinnitus.

Line 194: remove “we the” from “models, we the noise was applied”

Line 539: “an effect visible throughout the experiment (Figs 1A)” Figure 1A doesn’t show an effect just experimental design?

Line 642: “Although ABRs were not performed in sham-operated control cases”. I thought no sham controls were performed? All 7 animals were exposed to AT?

Line 694: “also showed reduced sleep disruption (3B)” Figure 3 B shows gap detection.

Legends for all figures should show n for each group. Please explain the colour coding. I also assume the symbols represent different animals. However there is an upside down triangle in fig 1B baseline that is not visible in 1 week?

Regarding Figure 1 E: what does the asterisk represent? Are the 3 groups different from each other under NBN condition? They seem identical and the difference not noteworthy.

It would be useful to show tinnitus index for all animals where available. Not just for the 3 animals that had sleep analysed. It seems all animals developed tinnitus? This is unexpected as other animals models show around 50% of animals to develop tinnitus, similar to humans where not everyone with a hearing loss develops tinnitus.

Figure 4 E and F do not show the 6 month data for case 2 and 3?

Discussion:

“we provided the first empirical evidence provided initial evidence supporting the idea that tinnitus emergence coincides with emergence of sleep disruption.” This is an overstatement. Tinnitus development was only monitored at two timepoints, long time apart.

With regards to tinnitus studies in humans it would be worthwhile to discuss whether these studies investigated people bothered by their tinnitus as their stress and anxiety levels are likely to affect sleep patterns.

“More specifically, elevated cortical activity has been reported in humans [81,82] and in a range of animal models of induced tinnitus (chinchillas: [80], cats: [83], guinea pigs: [84,85], rats: [86], and mice: [87]), including our own preliminary results (Fig S16).” See my previous comment re Meredith et al. It is also associated with hearing loss per se.

The increased hearing loss at 6 months is very surprising. Other animal models do not show this following an AT. Hearing thresholds are generally stable after 1 or 2 weeks. Hence statement “The main limitations from this case study are the absence of temporary shifts in ABRs following noise overexposure” is incorrect as immediate loss was not measured.

Conclusion section needs to be tempered as “states provided evidence for a bi–directional interaction between tinnitus and natural brain state dynamics” is an overstatement of the results.

7. PLOS authors have the option to publish the peer review history of their article (what does this mean?). If published, this will include your full peer review and any attached files.

Reviewer #1: No

Reviewer #3: **Yes: **Konstantin Tziridis

Reviewer #4: No

---

## [Author Response · Author response to Decision Letter 1]

1 Nov 2024

Oxford, November 01, 2024

Dear Dr Bruns, PLoS One Editor,

Re: PONE-D-24-17819R2 “Cortical evoked activity is modulated by the sleep state in a ferret model of tinnitus. A cross-case study.”

We are pleased that you are interested in publishing our study in the Journal. We have worked hard to address every point raised by the reviewers and we hope that the manuscript is now ready to be accepted in PLoS One.

We deeply apologize for the oversights on our part in submitting the previous revised version of the manuscript in which the last third of the references were missing. We have now revised the manuscript carefully to ensure that the reference list is complete.

Our Revised version 1 manuscript had 4 main figures and 16 supplementary figures, and no figures were missing. There were no new figures 5 and 6 in that version, but as Reviewer #3 noticed (see Rev.#3.10, but also Rev.#4.5), line 706 included an incorrect reference to Fig 6A-C instead of Fig. 2A-C, which has been corrected in the present submission.

In this revised version, we have improved the figures as requested by Reviewer #1, the readability of the introduction and discussion as requested by Reviewer #3, and the conclusions have been toned down (Reviewer #4) to match our results. 

In the cover letter and response to the reviewers, we use Arial 12 to distinguish our responses from the editorial and reviewer comments, which have been kept in the original font (Segoe Ul 11.5). Revisions made to the manuscript text are shown in red. 

With this letter, we submit a rebuttal letter with the response to the reviewers, a marked-up copy of our manuscript that shows in red changes made to the original version 'Revised Manuscript 2 with Track Changes', an unmarked version of our revised paper without tracked changes labelled 'Manuscript2', and a new set of main figures and supplementary figures.

Kind regards,

Victoria M Bajo

Reviewers' comments:

Reviewer's Responses to Questions

Comments to the Author

1. If the authors have adequately addressed your comments raised in a previous round of review and you feel that this manuscript is now acceptable for publication, you may indicate that here to bypass the “Comments to the Author” section, enter your conflict of interest statement in the “Confidential to Editor” section, and submit your "Accept" recommendation.

Reviewer #1: (No Response)

Reviewer #3: (No Response)

Reviewer #4: (No Response)

2. Is the manuscript technically sound, and do the data support the conclusions?

Reviewer #1: Yes

Reviewer #3: Yes

Reviewer #4: Partly

3. Has the statistical analysis been performed appropriately and rigorously?

Reviewer #1: N/A

Reviewer #3: Yes

Reviewer #4: Yes

4. Have the authors made all data underlying the findings in their manuscript fully available?

Reviewer #1: Yes

Reviewer #3: Yes

Reviewer #4: Yes

5. Is the manuscript presented in an intelligible fashion and written in standard English?

Reviewer #1: Yes

Reviewer #3: Yes

Reviewer #4: Yes

6. Review Comments to the Author

Reviewer #1: I thank the authors for addressing my comments and adding valuable data to the manuscript as well as editing the text for clarity. I have a few remaining suggestions, the overall comment remaining being that the manuscript would benefit greatly from another round of editing on the figures.

-The manuscript can be hard to digest given that it explores the broad topics of tinnitus and sleep, and as a case study requires the reader to consider each individual rather than a big-picture result. As the authors chose to keep the paper together rather than splitting it as R2 suggested, to improve readability I would suggest moving a couple items from the supplemental material to the main figures:

-One of the main results highlighted in the discussion, that is consistent across the 3 animals -- an increase in REM2-- is not explicitly shown in a figure (the REM2 amounts are shown in separate figures for the baseline and post-NOE conditions). Pre- and post- NOE REM2 measurements (along with the amounts of other sleep stages) should instead be side-by-side in a figure, with error bars, and statistical analysis if possible.

Rev #1.1. We have added a new Fig. 5, which includes the panels from the original Fig. S5 along with additional panels (Fig. 5E) that explicitly show the average changes in REM2 over time (baseline, 1 week, and 6 months, presented side-by-side), as well as changes in Wake, REM and NREM amounts. Since only three animals contributed to these plots, we represent individual animals with thin grey lines instead of using error bars. Due to the small sample size, we did not conduct a statistical analysis. However, we agree with the reviewer that this presentation more effectively illustrates the transient increase in REM2 across all animals following NOE.

-The behavioral schematics from figure s1 should be moved to figure 1 as this adds valuable context.

Rev #1.2. We have modified Figure 1 moving panels A, B, and D from Supplementary 1 to Figure 1, and changes in the main text were made accordingly. To avoid including too many panels in Fig.1, we moved old Fig.1 C, D and F into Fig. S1, now titled: Assessment of auditory function in the ferret model: baseline assessments and long-term performance in operant gap detection.

-In general, the figures should be edited for visibility. In places the font sizes are different across panels, e.g. figures 3 and 4. In Figure 3B, middle, there are some graphics that appear to be a screenshot artifact that should be removed.

Rev #1.3. Figures have been further edited, artefacts (for example in Figure 3B) have been removed, and font sizes across panels and axes have been checked and updated as necessary. Specifically: axis labels in Fig. 3A, Fig. 4G and in the insets in Fig. 3C have been altered, and minor changes have been made in supplementary Figures S2-6, 10 and 15.

-The components that make up the tinnitus index calculation, and the assumptions of the TI, should be described verbally in the results section, as this gives important context to the data shown in figures.

Rev #1.4. As requested, the components of the tinnitus index calculation and the assumptions of the TI calculation have been described and summarised in the results section. 

Line 556: “The tinnitus index was calculated as the sum of three metrics obtained by comparing the values before and after the noise overexposure, with positive values indicating changes in line with tinnitus development (see Methods). 

Eq(1): TI=M(cont)+M(thresh)+ M(silence)

The metrics for operant gap detection performance refer to the change in the ability of an animal to detect the continuous sound (no gap) across all tested stimulus frequencies and to the change in gap-detection threshold across all stimuli (Eq(2) and Eq(3) in the Methods). The metric for operant silence detection refers to the change in silence detection ability (percent correct score in silence trials) after noise overexposure (Eq(4) in Methods).”

-In the discussion, lines 897-898 contain a typo.

Rev #1.5. Deleted “The first scientific evidence”

Reviewer #3: The authors present behavioral and electrophysiological data investigating the link between tinnitus and sleep in 7 adult female ferrets, with 3 animals receiving long term EEG recordings and from 2 (+1) animals, auditory cortex activity was recorded with sharp electrodes. This resulted in three single cases, that could be described over six months in great detail. The manuscript is already revised and most concerns regarding the methodology and results have been accounted for.

Overall I suggest a major revision, especially in the Introduction and Discussion sections, in the latter many references (after 68) are missing.

Major points:

One major point that I am missing from the manuscript is the introduction to the different neurophysiological models of tinnitus development and the introduction to sleep and sleep related measurements (polysomnography). Both are relevant for the understanding of the used methods and the interpretation of the results.

Rev #3.1. The introduction section has been edited by adding a section on the main neurophysiological models of tinnitus development and an introduction to sleep and polysomnography.

Line 59: “Three main models have been implicated in the development of tinnitus, which attempt to explain the changes that occur in the brain following a cochlear insult: altered lateral inhibition, homeostatic plasticity, and stochastic resonance (reviewed in [19]). In the lateral inhibition model, tinnitus is caused by disinhibition of frequency channels adjacent to deafferented channels (edge effect; [20]); in the homeostatic plasticity model by an increase in spontaneous activity due to an increase in central neuronal gain [21], and, in the stochastic resonance model, by increased noise in a recurrent neural network, leading to subthreshold auditory signals becoming detectable [19].

Tinnitus is associated with functional changes in widely distributed brain regions, including both auditory and non–auditory areas [22–28], many of which exhibit a dramatic modulation in their spatiotemporal activity across vigilance states: for example, wakefulness is dominated by high-frequency, low-amplitude oscillatory brain activity, whereas sleep is dominated by slower, high-amplitude oscillations [29–31].”

Line 93: “We tracked behavioural markers of noise–induced tinnitus and hearing impairment in a ferret model at one week and six months after noise overexposure and, in parallel, assessed the sleep–wake pattern, as well as spontaneous and auditory–evoked EEG activity across vigilance states. In a similar manner to polysomnography, which is based on electromyogram (EMG) recordings and EEG recordings of global brain activity to assess human sleep and sleep disorders [4,29,40], we used EEG and EMG recordings for vigilance state scoring as previously established in rodents (e.g. [14,41]) and ferrets [42].”

For the reader, it would improve the readability, if instead of post1 and post2 the real timepoints are presented in the text and Figures. In some of the supplements, this is already the case and improves the intelligibility significantly.

Rev #3.2. This is a good idea. We now refer in the text and figures to the Post1 and Post2 timepoints as ‘1 week’ and ‘6 months’. 

L888ff: Add here also the development aspect from the introduction. This is relevant, as you have early and late recording timepoints. Also compare your behavioral model of tinnitus identification to other behavioral animal models, including conditioning and reflexive models.

Rev #3.3. The Discussion has been edited to include the different models of tinnitus development and a comparison of our behavioural methods of tinnitus identification (operant gap and silence detection) with other reflexive models. 

Line 1004: “Cochlear damage after noise overexposure can lead to a compensatory increase in neuronal excitability that restores evoked activity. This can occur through mechanisms such as homeostatic plasticity [21] or adaptive stochastic resonance [87], processes that can themselves drive tinnitus development [19,21,87]. Elevated cortical activity has been reported in human tinnitus sufferers [88,89] and in a range of animal models of induced tinnitus (chinchillas: [86], cats: [90], guinea pigs: [91,92], gerbils [93,94], rats: [95], and mice: [96]), including our own preliminary results (Fig S15). 

The increased evoked activity we observed in the first assessment after NOE could be explained by adaptive stochastic resonance, which operates on short timescales [19]. In contrast, sustained tinnitus six months after NOE, especially in the context of higher levels of hearing loss, is more likely based on plastic changes [97] relating to homeostatic plasticity and alterations in central gain [21].

Assessments of tinnitus in our study are based on positive operant behavioural tasks rather than reflexive behaviours like the acoustic startle reflex, which show habituation that is particularly rapid in carnivores and primates, including humans [98]. However, there are further important differences in tinnitus identification based on operant and reflexive paradigms: for example, reflexive paradigms, such as classical assessment of the gap-induced inhibition of the behavioural startle response (GPIAS, [99,100]) make it challenging to separate tinnitus from hearing loss, whereas operant paradigms can be more sensitive to differences between these conditions (e.g. [55]). On the other hand, operant paradigms have a learning component that may affect the results (reviewed in [101]). For this reason, we completed extensive baseline training in all operant paradigms before testing animals for tinnitus (see methods).

Please check for the missing references after 68. Please also increase your comparisons of ABR shifts to more than one reference (69) for animals and humans (70).

For all references after 68, I cannot decide if they are appropriate or missing points. Please double-check. 

Rev #3.4. We deeply apologize for the oversight on our part in submitting the previous revised version of the manuscript with the last third of the references missing. We can confirm that the reference list is complete. 

To avoid including a long list of citations, we now use two reviews (Shore et al., 2016 and Baguley, 2002) that describe the variable effects of noise trauma on tinnitus and hearing impairment in experimental animals and the tinnitus heterogeneity in humans, respectively (Lines 964-967). 

Lines 964-967: “Interestingly, we show that the measures of the degree of tinnitus or its perception by the animal, as well as hearing impairment following noise overexposure are highly idiosyncratic, in line with variable effects of noise trauma seen in other animal models and the tinnitus heterogeneity that is characteristically found in humans (for example reviewed in [76] and [77], respectively).”

L 955: An important tinnitus animal model is the Mongolian gerbil, please add references here.

Rev #3.5. References added to Line 1009 [Tziridis et al., 2015; Jeschke et al., 2021] and to the reference list.

Tziridis K, Ahlf S, Jeschke M, Happel MF, Ohl FW, Schulze H. Noise Trauma Induced Neural Plasticity Throughout the Auditory System of Mongolian Gerbils: Differences between Tinnitus Developing and Non-Developing Animals. Front Neurol. 2015 Feb 10;6:22. doi: 10.3389/fneur.2015.00022. 

Jeschke M, Happel MFK, Tz

---

## [Decision Letter · Decision Letter 2]

14 Nov 2024

Cortical evoked activity is modulated by the sleep state in a ferret model of tinnitus. A cross-case study.

PONE-D-24-17819R2

Dear Dr. Bajo,

We’re pleased to inform you that your manuscript has been judged scientifically suitable for publication and will be formally accepted for publication once it meets all outstanding technical requirements.

Kind regards,

Patrick Bruns

Academic Editor

PLOS ONE

Additional Editor Comments (optional):

Reviewers' comments:

Reviewer's Responses to Questions

**Comments to the Author**

1. If the authors have adequately addressed your comments raised in a previous round of review and you feel that this manuscript is now acceptable for publication, you may indicate that here to bypass the “Comments to the Author” section, enter your conflict of interest statement in the “Confidential to Editor” section, and submit your "Accept" recommendation.

Reviewer #3: All comments have been addressed

Reviewer #4: All comments have been addressed

2. Is the manuscript technically sound, and do the data support the conclusions?

Reviewer #3: Yes

Reviewer #4: Yes

3. Has the statistical analysis been performed appropriately and rigorously? 

Reviewer #3: Yes

Reviewer #4: Yes

4. Have the authors made all data underlying the findings in their manuscript fully available?

Reviewer #3: Yes

Reviewer #4: Yes

5. Is the manuscript presented in an intelligible fashion and written in standard English?

Reviewer #3: Yes

Reviewer #4: Yes

6. Review Comments to the Author

Reviewer #3: Thank you for putting so much efford in improving your manuscript.

I have no kore comments to make to your work.

Reviewer #4: all comments have been addressed by the authors. manuscript reads well and is suitable for publication.

7. PLOS authors have the option to publish the peer review history of their article (what does this mean?). If published, this will include your full peer review and any attached files.

Reviewer #3: **Yes: **Konstantin Tziridis

Reviewer #4: No

---

## [Editor Report · Acceptance letter]

22 Nov 2024

PONE-D-24-17819R2 

PLOS ONE

Dear Dr. Bajo, 

I'm pleased to inform you that your manuscript has been deemed suitable for publication in PLOS ONE. Congratulations! Your manuscript is now being handed over to our production team.

Kind regards, 

on behalf of

Dr. Patrick Bruns 

Academic Editor

PLOS ONE